# Nageotte nodules in human dorsal root ganglia reveal neurodegeneration in diabetic peripheral neuropathy

Stephanie I. Shiers [1] ✉, Khadijah Mazhar[1], Andi Wangzhou[1], Rainer Haberberger [2], Joseph B. Lesnak [1], Nwasinachi A. Ezeji[1], Ishwarya Sankaranarayanan[1], Diana Tavares-Ferreira [1], Anna Cervantes[3], Geoffrey Funk [3], Peter Horton[3], Erin Vines[3], Gregory Dussor[1] & Theodore J. Price [1] ✉

Nageotte nodules, first described in 1922 by Jean Nageotte, are clusters of non-neuronal cells that form after sensory neuron death. Despite their historical recognition, little is known about their molecular identity nor their involvement in neuropathies that involve neuronal loss like diabetic peripheral neuropathy (DPN). In this study, we molecularly characterize Nageotte nodules in dorsal root ganglia recovered from organ donors with DPN. Here we show that Nageotte nodules are abundant in DPN sensory ganglia and account for 25% of all neurons. Peripherin-and Nav1.7-positive dystrophic axons invade Nageotte nodules, forming small neuroma-like structures. Using histology and spatial sequencing, we demonstrate that Nageotte nodules are mainly composed of satellite glia and non-myelinating Schwann cells that express *SPP1* and are intertwined with sprouting sensory axons originating from neighboring neurons. Our findings suggest that Nageotte nodules are an integral feature of dorsal root ganglion neurodegeneration, providing potential therapeutic targets for sensory neuron protection and pain management in DPN.

Jean Nageotte, a French neuroanatomist, first described residual nodules, now known as Nageotte nodules, in 1922 after conducting grafted ganglia experiments in rabbits. He described these structures as clusters of satellite glia that fill the space of decomposed sensory neurons: "As the nerve cell corpse is reabsorbed, the satellite cells proliferate, and when the nerve cell has disappeared, they form a nodule[1]." Since his discovery, Nageotte nodules have been infrequently described in the literature with the majority of papers reporting their presence in ganglia from macaques (4 publications;[2–5]), humans (21 publications;[6–26]), and rats (3 publications[27–29]) associated with various neurodegenerative conditions and rare diseases. Human data on Nageotte nodules are limited to neuropathological findings in sensory ganglia with long post-mortem intervals (PMIs) and are, in most cases,

from single case studies. As such, little is known about the molecular identity of these structures, nor their involvement in the context of pain and neurodegeneration. The identity and potential clinical importance of these pathological structures appear to be a story that has been lost to time.

Diabetic peripheral neuropathy (DPN) is the most common form of neuropathic pain. Patients often report spontaneous shooting or stabbing pain accompanied by sensory deficits, which is usually attributed to die-back of sensory axons from the epidermis[30]. Sural nerve biopsies show that this axonal loss in DPN can be extensive. The stabbing and shooting pain in DPN is attributed to spontaneous action potentials generated in nociceptors because microneurography studies in patients show that pain is correlated with activity in these

[1]Department of Neuroscience, Center for Advanced Pain Studies, The University of Texas at Dallas, Richardson, TX, USA. [2]Anatomy and Pathology, The University of Adelaide, Adelaide, SA, Australia. [3]Southwest Transplant Alliance, Dallas, TX, USA. ✉e-mail: stephanie.shiers@utdallas.edu; theodore.price@utdallas.edu

axons[31–34]. While DPN is typically considered an axonal disease, structural abnormalities also occur in the dorsal root ganglia (DRGs) such as neuronal loss[35] and the formation of dystrophic axonal swellings[26,36]. While Nageotte nodules form as a result of neuronal death, only two studies have reported Nageotte nodules in diabetes: one, a case study of an individual with fulminant type 1 diabetes[17], and two, a single individual with DPN (out of 5 that were investigated) in a pathology study from 1964[37]. It is important to note that these tissues were acquired after autopsy and displayed extensive vacuolization of cellular compartments which is now known to be an artifact of prolonged post-mortem interval[36] and/or improper tissue freezing[38]. It is still unknown if Nageotte nodule formation accompanies ganglionic cell loss in DPN, and if these structures play a role in the pathogenesis of pain and sensory loss in the disease. Uncovering the molecular identity of Nageotte nodules may unlock new mechanistic insights into DRG neurodegeneration and reveal new neuroprotective treatment options for diabetic sensory neuron loss or for the treatment of diabetic pain.

In this work, we combined Jean Nageotte's historical notes with modern-day technologies (confocal imaging/spatial sequencing) to molecularly characterize Nageotte nodules. We screened DRGs from 90 organ donors and found numerous Nageotte nodules in DRGs recovered from individuals with DPN and other types of neuropathies. Intertwined with Nageotte nodules, we found a neuroma-like axon bundle which has not been described before in contemporary

literature, nor in any animal model of diabetes. These axonal arborizations express sensory markers like peripherin, Nav1.7, and TrpV1, but not the sympathetic marker, tyrosine hydroxylase. Using axonal tracing, we find that surviving DPN sensory neurons lose their pseudounipolar morphology and sprout multiple neurites from their glomeruli and cell body, forming pericellular nests and Nageotte nodule axonal bundles. Using spatial transcriptomics and histological validation, we identify that non-myelinating Schwann cells and satellite glia form Nageotte nodules and express secreted ligands like osteopontin that could interact with receptors like CD44 on surviving neurons. These ligand-receptor interactions between Nageotte nodules and sensory neurons may be key to understanding how to treat DPN.

## Results

### Nageotte nodules form in DRGs in DPN and other types of neuropathies

To investigate the molecular identity of Nageotte nodules and to address their role in neuropathy, we recovered DRGs from a large sample of organ donors (90 donors) with short PMIs (average: 2 h) (Supplementary Data 1). Tissue quality was assessed using established protocols[38]. We first conducted Hematoxylin and Eosin staining on all DRGs (90 donors) to visualize Nageotte nodules which appear as dense clusters of non-neuronal nuclei (Fig. 1A). Each DRG was qualitatively scored for the presence of Nageotte nodules using a 5-point scaling

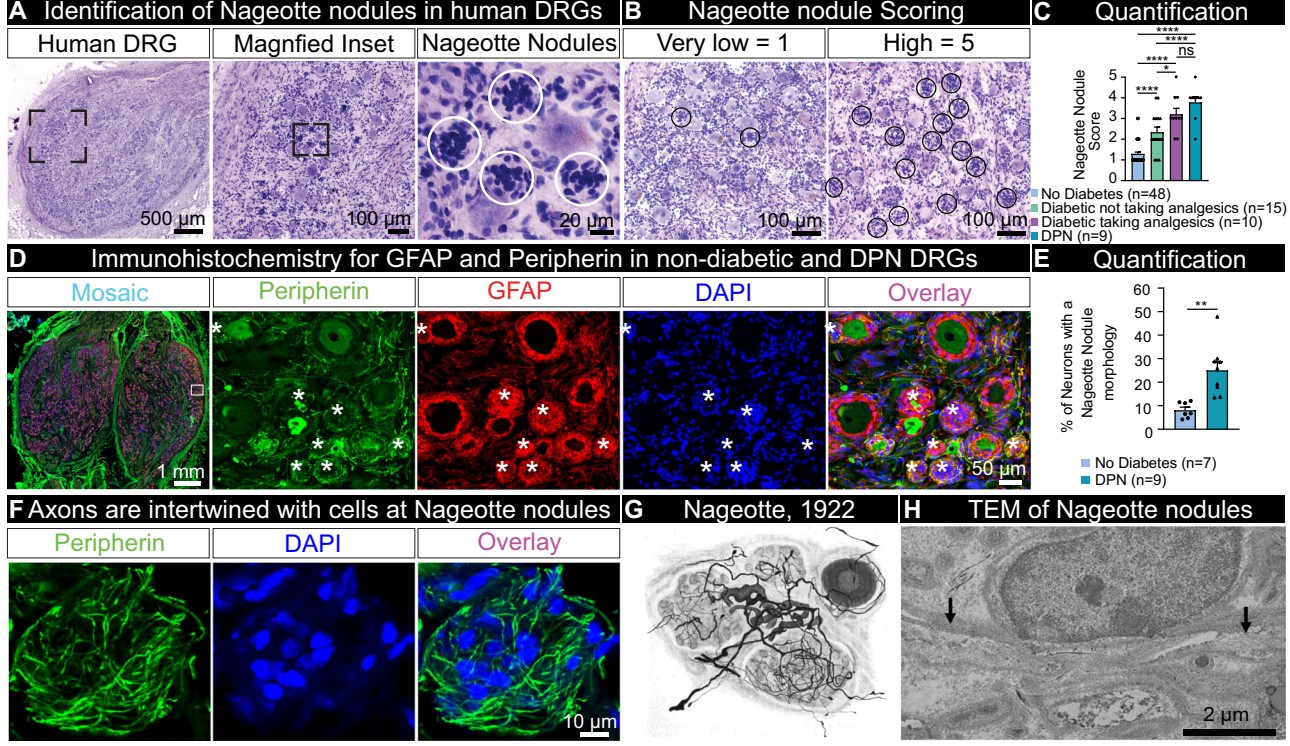

**Fig. 1 | Identification of Nageotte nodules in DRGs from diabetic organ donors. A** Hematoxylin and eosin staining was performed on DRGs from organ donors (*n* = 90), **B** and then each DRG was scored for prevalence of Nageotte nodules using a qualitative scoring system. **C** After scoring, donors were grouped based on their medical history of diabetes, analgesic usage, or medical note of diabetic peripheral neuropathy (DPN). Diabetic donors had significantly higher Nageotte nodule scores compared to non-diabetic donors without pre-existing pain conditions. In diabetics, Nageotte nodule content increased in severity in relation to DPN as indicated by analgesic usage, medical note of DPN, and/or diabetes-related amputation. **D** Representative images of an L4 bi-ganglia from a DPN donor immunostained for GFAP (red, satellite glial cells and non-myelinating Schwann cells), peripherin (green, sensory neurons), and DAPI (blue, nuclei). Asterisks denote Nageotte nodules. Sample size: Non-diabetic *n* = 7; DPN *n* = 9. **E** The

percentage of Nageotte nodules was significantly higher in the DPN DRGs (average: 25%) compared to non-diabetic DRGs (average: 8%). **F** Confocal image of a peripherin-positive axon bundle intertwined with other cells at Nageotte nodules. Similar staining was observed across all 9 DPN donors. **G** Image taken from Jean Nageotte's original 1922 publication[1] in which three Nageotte nodules (mid top, mid bottom, and left) contain axon bundles which sprout from a glomerulus (middle). **H)** Transmission electron microscopy (TEM) of a Nageotte nodule. Arrows point to unmyelinated axonal fibers. Sample size: Diabetic taking analgesics *n* = 2. Statistical tests: (**C**): One-way ANOVA with Bonferroni's multiple comparisons test. *$p$ = 0.0309, ****$p$ < 0.0001, ns (not-significant) $p$ = 0.5525. **E**: Unpaired two-sided *t*-test. **$p$ = 0.0013. Data points represent individual donors. Error bars = mean +/- SEM. Scale bars: (**A**): 500 µm, 100 µm. 20 µm (**B**): 100 µm. (**D**): (Mosaic−1 mm and other panels−50 µm. (**F**): 10 µm. (**H**): 2 µm.

system ranked from very low-to-high (Fig. 1B). Following scoring, donors were categorized into groups based on keywords in their medical histories (bolded in Supplementary Data 1). The "diabetic with peripheral neuropathy" group included diabetic donors with documented neuropathy, nerve pain, and/or amputation. The "diabetic taking analgesics" group consisted of diabetic donors without neuropathy-related keywords but who were prescribed analgesics (gabapentin, duloxetine, hydrocodone, etc). The "diabetic not taking analgesics" group includes diabetics with no neuropathy-related keywords and no prescribed analgesics. The "no diabetes" group consisted of non-diabetic donors without DRG-related chronic pain conditions such as arthritis, neuropathy, or fibromyalgia. The results showed that Nageotte nodules were significantly more prevalent in diabetics with peripheral neuropathy, diabetics taking analgesics, and diabetics not taking analgesics compared to non-diabetic controls (Fig. 1C), respectively.

It is important to note that the donor medical histories are provided by the organ procurement organization (OPO) and are summaries of hospital records and family-reported information. In some cases, neuropathy or pain may have been undiagnosed or unreported by the donor's family. For example, donor #12 who had high Nageotte nodule content, was taking gabapentin and duloxetine (two widely prescribed analgesics for diabetic nerve pain), and had difficulty walking for the past 6 weeks, fell under the category of "diabetic taking analgesics." However, this donor likely had neuropathy, but it was not explicitly documented in the medical history summary, so it did not meet the criteria for being grouped as "diabetic with peripheral neuropathy."

A small subset of non-diabetic donors with other pain conditions such as fibromyalgia or idiopathic peripheral neuropathy also showed high numbers of Nageotte nodules within DRGs (Supplementary Fig. 1A, B). This appeared to be specific for pain disorders with a neuropathic component because Nageotte nodules were not routinely noted in DRGs obtained from donors with arthritis. However, the affected arthritic joints were not noted in the medical histories, thereby, the DRGs associated to arthritic pain dermatomes may not have been investigated. Lumbar (L5) and sacral (S1) DRGs recovered from a diabetic donor with below-the-knee amputation of the right leg revealed Nageotte nodules in DRGs from both sides of the body (Supplementary Fig. 1C), supporting that Nageotte nodule formation is not unilateral, nor a result of amputation. In a subset of DPN donors, we were also able to procure DRGs from the upper thoracic area (T4). We observed comparable Nageotte nodule content between lumbar and thoracic 4 DRGs in the DPN donors (Supplementary Fig. 1D, E), indicating that Nageotte nodule formation also likely occurs across DRG levels/dermatomes.

## Neuroma-like axon bundles intertwine with the cells at Nageotte nodules

Jean Nageotte reported that the cells comprising Nageotte nodules are satellite glial cells (SGCs)[1]. In order to quantify the percentage of Nageotte nodules in relation to the total neuronal population, we conducted immunohistochemistry for glial fibrillary acidic protein (GFAP), a marker of SGCs and non-myelinating Schwann cells[39], and peripherin, a sensory neuron marker, in non-diabetic and DPN DRGs. Nageotte nodules can be identified using nuclear stains[6–8,11,13,17,18,24,40], and we found robust GFAP signal localized to the cells at Nageotte nodules (Fig. 1D). When quantified, we found that 25% of the neurons had a Nageotte nodule morphology in the DPN DRGs, suggesting that a quarter of all sensory neurons are dead in these individuals (Fig. 1E). To identify what population of neurons was dying, we first measured the diameter of neurons and observed a significant reduction in the percentage of small diameter neurons (<65 µm in human[41,42]) in the DPN DRGs compared to non-diabetic controls (Supplementary Fig. 2A, B). These results corroborated RNAscope expression data in which the

nociceptor (*SCN10A*+) population significantly decreased in DPN DRGs (Supplementary Fig. 2C, D).

Peripherin staining revealed a neuroma-like axonal structure intertwined with the cells forming Nageotte nodules (Fig. 1F) similar to descriptions by Jean Nageotte in rabbit DRG in 1922[1] (Fig. 1G). Nageotte posited that these "arborizations of residual nodules" were axonal sprouts from the hypertrophied glomeruli of surviving neurons: "extremely rich bouquets of fibers, which arise from the glomeruli of surviving nerve cells, and which will flourish in the neighboring residual nodules, formed by the satellite elements of dead nerve cells"[1] (Fig. 1G). Transmission electron microscopy supported the existence of thin, unmyelinated fibers intertwined with the cells at Nageotte nodules (Fig. 1H, Supplementary Fig. 3).

## Extensive axonal sprouting and pericellular nest formation in DPN DRGs

Interestingly, there was also a significant increase in peripherin-positive axonal sprouting that spanned the entirety of the DPN DRGs compared to non-diabetic controls (Supplementary Fig. 4A, B). These axonal sprouts not only intertwined with the cells forming Nageotte nodules but also surrounded sensory neurons with visible cell bodies (Supplementary Fig. 4C, Supplementary Movie 1). In rodents and humans with neuropathic pain, tyrosine hydroxylase (TH)-expressing sympathetic axons are known to encircle sensory neurons, forming basket-like structures called pericellular nests (PCNs)[43–45]. Nageotte nodules are anatomically distinguishable from PCNs because they do not have a neuronal soma and are conglomerates of non-neuronal cells. However, we also observed numerous PCNs in the DPN DRGs (Supplementary Fig. 4C). Jean Nageotte surmised that the axons comprising Nageotte nodules and PCNs were not sympathetic in origin as they formed within 24 h in his ganglia preparations, were highly numerous, and sprouted from sensory neurons[1] (Supplementary Fig. 4D). He also postulated that the PCNs and Nageotte nodule arborizations were the same structures at difference stages of neuron decay/death (Supplementary Fig. 4D, E): "These arborizations of residual nodules and Dogiel's pericellular platoons are one and the same thing. I was able to convince myself that, at the beginning, all the arborizations of the residual nodules begin as pericellular platoons developed around dying or dead nerve cells[1]."

## Nageotte nodule arborizations do not express TH

To assess possible sympathetic sprouting in DPN DRGs, we labeled sympathetic fibers using tyrosine hydroxylase (TH) which robustly stained sympathetic neurons and fibers in the human sympathetic chain ganglion (Supplementary Fig. 5A) and sparse fibers within the nerve attached to the DRG (Supplementary Fig. 5B) but showed little-to-no axonal labeling within the DRG (Supplementary Fig. 5C). We found no evidence for TH-positive axons at Nageotte nodules (Fig. 2A), nor at PCNs, indicating that these arborizations are not sympathetic. It is important to note that the high population of non-sympathetic PCNs in the DPN DRGs is largely contrasting to the prevalence of sympathetic PCNs noted in rodents and humans with other types of neuropathic pain[43–45]. For example, only 2 sympathetic PCNs were found in a DRG from a human with herniated intervertebral disc and severe sciatica pain[45]. The differences in prevalence and in the expression of TH of these morphologically similar structures indicate that these are likely two distinct pathologies.

## Nageotte nodule arborizations and dystrophic axons express TrpV1 and Nav1.7

We next sought to identify the nature of these sprouting axons into Nageotte nodules, hypothesizing that they could be nociceptive fibers. To test this hypothesis, we conducted immunohistochemistry for the

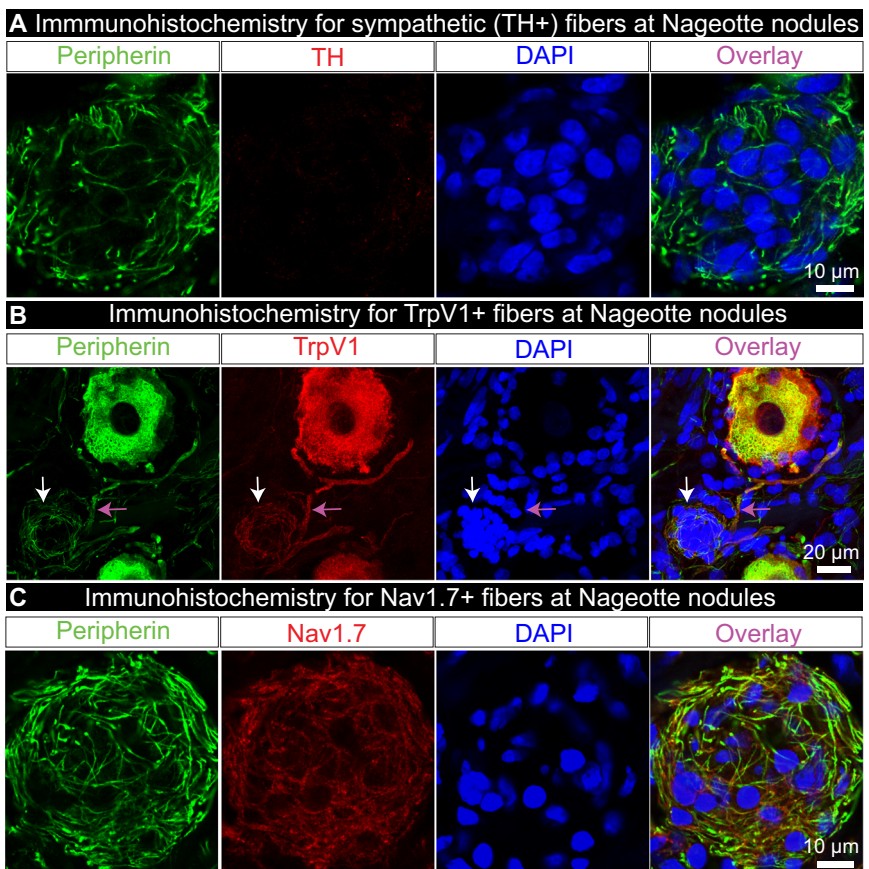

**Fig. 2 | Nageotte nodule axons express nociceptive, but not sympathetic markers. A** Tyrosine hydroxylase (TH, red) labeling in combination with peripherin (green) and DAPI (blue) revealed that Nageotte nodule axonal sprouts were not sympathetic in origin. Sample size: DPN $n = 9$. **B** TrpV1 fibers at a Nageotte nodule in a DPN DRG (white arrow). TrpV1 was only detected in Nageotte nodules from a single DPN donor out of 5 that were investigated. The TrpV1+ fibers at the Nageotte nodule appeared to arise from a glomerulus (magenta arrow). Sample size: DPN $n = 5$. **C** Nav1.7 (red) was detected in the axonal fibers intertwined with Nageotte nodules. Sample size: DPN $n = 6$. Scale bars: **A**: 10 µm. **B**: 20 µm. **C**: 10 µm.

capsaicin-receptor, TrpV1, which is expressed at the mRNA and protein level in all human nociceptors[41,42,46]. However, the majority of the DPN samples, particularly the DRGs obtained from donors who had established diagnoses of diabetic peripheral neuropathy or diabetes-related amputation, showed a drastic decrease in TrpV1 protein expression throughout the DRG (Supplementary Fig. 6A, B). These data corroborated published mRNA sequencing data in which TrpV1 mRNA is significantly decreased in human DPN DRGs[35]. However, one DPN DRG from an organ donor who was diabetic, taking gabapentin and duloxetine, and had difficulty walking (donor 12) showed elevated TrpV1 expression within the DRG (Supplementary Fig. 6A, B). In this donor, TrpV1 was detected in axons at Nageotte nodules (Fig. 2B), and in dystrophic axons (Supplementary Fig. 6C). TrpV1-positive axonal fibers at Nageotte nodules arose from a thickened portion of a TrpV1-positive axon, potentially a glomerulus given its close proximity to two TrpV1-positive sensory neurons (Fig. 2B).

We then assessed Nav1.7 expression in axonal arborizations at Nageotte nodules. In humans, the voltage gated sodium channel (VGSC) Nav1.7 is known to be expressed in sensory neurons, including all human nociceptors[41,46], and in painful neuromas (41–43) which are suspected to give rise to ectopic activity as has been evidenced by the efficacy of VGSC blockers in experimental (44–47) and human neuromas (48). Similar to neuromas, Nageotte nodules express regenerative axon markers like GAP-43 (26) and morphologically appear similar given the abundance of abnormal axonal sprouting. Nav1.7 was expressed by the axonal arborizations at Nageotte nodules (Fig. 2C, Supplementary Fig. 7), in axonal fibers

throughout the DRG including PCNs, and within the membrane of dystrophic axons which were frequently observed in the DPN DRGs (Supplementary Fig. 8A, B). Neuroaxonal dystrophy is a known pathology in humans with diabetes[26,36], and is marked by the formation of dystrophic axons that are believed to form as a result of "frustrated axonal regeneration" in which the axon terminals of regenerating sensory fibers swell, and contain large numbers of neurofilaments, vesicles, and neuropeptides like CGRP[26]. Jean Nageotte reported similar structures in decaying spinal ganglia which he called "adventitious buds" or "growth balls." He claimed these structures sprouted from the sensory neuron soma, glomerulus, and the extracapsular portion of the axon in a process that he called collateral regeneration (Supplementary Fig. 8B–D).

## Nageotte nodule arborizations originate from nearby sensory neurons

Jean Nageotte's theory of collateral regeneration contrasts with the classically defined pseudounipolar shape of sensory neurons as it suggests that sensory neurons can take on a multipolar phenotype in degenerating DPN DRGs. We describe two lines of evidence supporting the hypothesis that DRG neurons take on a multipolar phenotype when they sprout into Nageotte nodules. First, we conducted filament tracing of peripherin-positive axonal fibers within DRG tissue sections to identify the origin of Nageotte nodule axon bundles. In many cases, Nageotte nodule axon bundles originated from discontinuous collaterals from outside the field of view; however, we were able to confidently trace the fibers in two Nageotte nodules which were

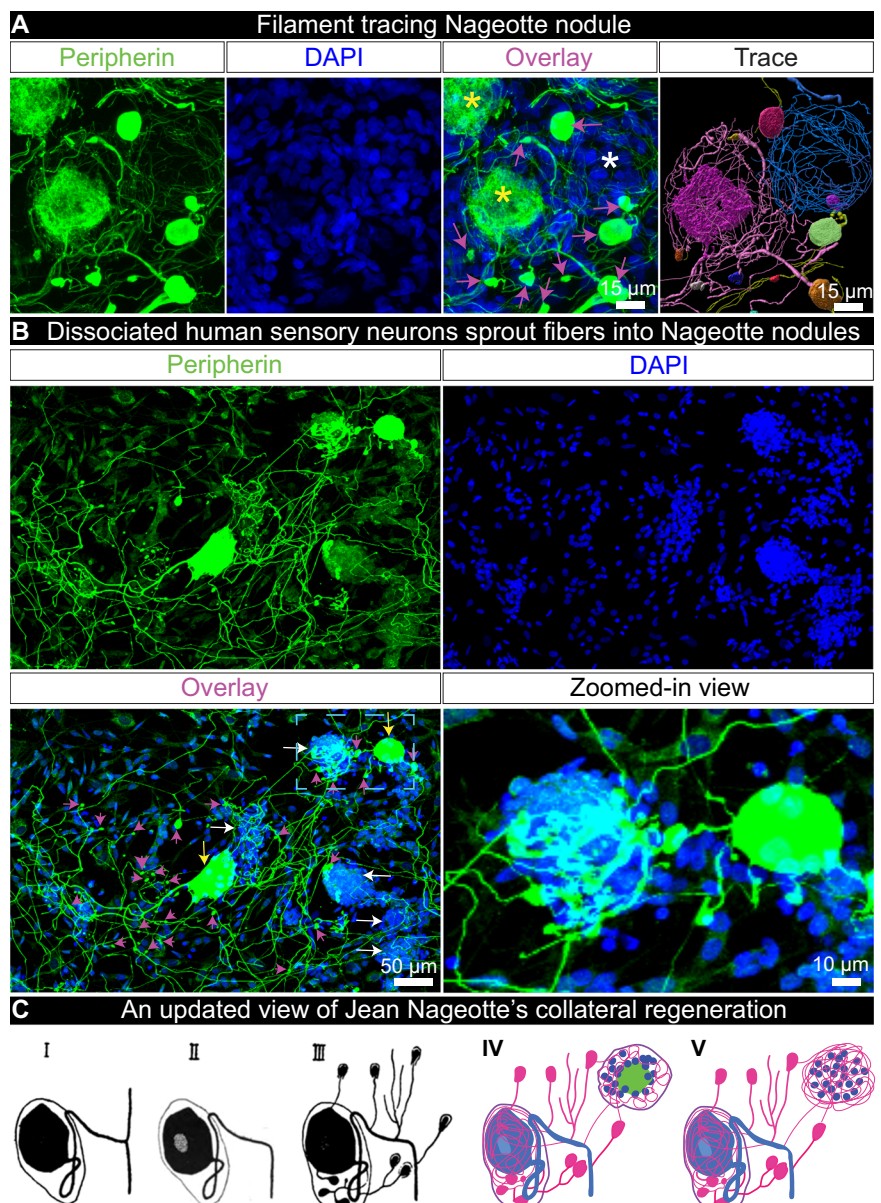

**Fig. 3 | Nageotte nodule axon bundles originate from axonal sprouts from local sensory neurons in situ and in vitro. A** Nageotte nodule (white asterisks), two sensory neuron cell bodies (yellow asterisks), and dystrophic axons (magenta arrows) in a 60X z-stack projection image of a DRG section stained for peripherin (green) and DAPI (blue) from donor 6. In the Trace panel, axonal filaments stemming from the neuronal soma of the middle sensory neuron were traced in Imaris (pink filament trace) which mainly connected to dystrophic axons (multi-colored axonal blebs). The Nageotte nodule axonal bundle (blue filament trace) was mainly spooling fibers that traced back to dystrophic axons. In some cases, dystrophic axons were interconnected to one another (yellow filament traces). Sample size: DPN with highest Nageotte score *n* = 1. **B** A representative 20X z-stack projection image of human DRG sensory neurons that were cultured in vitro for 3 days and then stained for peripherin (green) and DAPI (blue). Sample size: Non-diabetic *n* = 2. Human sensory neurons display a multipolar phenotype in which multiple axonal

branches sprout from the neuronal soma (yellow arrows), form dystrophic axons (magenta arrows), and intertwine with structures resembling Nageotte nodules (white arrows). A digitally zoomed-in view of the outlined area (cyan) exemplifies a multipolar sensory neuron sprouting fibers into a Nageotte nodule. **C** Jean Nageotte described collateral sprouting in 1922 (I–III; original illustration by Jean Nageotte, 1922) in which I) a normal ganglion cell with a T-bifurcated axon is II) deprived of the radicular branch of the axon resulting in III) neurite outgrowth from the soma and glomerulus which are equipped with encapsulated growth balls. Jean Nageotte as well as our imaging of DPN DRGs noted IV) the formation of non-sympathetic pericellular nests that formed around sensory neurons with intact somata and those with shrunken/misshapen somata likely in the process of dying. V) Neurites sprout from the dystrophic axons forming arborizations at Nageotte nodules. Scale bars: **A**: 15 µm. **B**: 50 µm and zoomed-in view panel: 10 µm.

connected to neurites and local dystrophic axons that budded from local sensory neurons (Fig. 3A, Supplementary Movie 2). We also traced many thin fibers that appeared to stem from the neuronal soma (Fig. 3A), forming a PCN around the same neuron and supporting not only a multipolar phenotype but also lending credence to Jean Nageotte's claim that non-sympathetic PCNs arise from sensory neurons. Second, dissociated sensory neurons from human DRGs take on a

multipolar morphology in vitro and display dystrophic axonal budding from the neuronal soma and elongated axonal branches that intertwine with in vitro structures that resemble Nageotte nodules (Fig. 3B, Supplementary Movie 3). Together, these findings support Jean Nageotte's original ideas on PCNs and Nageotte nodules and offer an updated view on Jean Nageotte's theory of collateral regeneration (Fig. 3C).

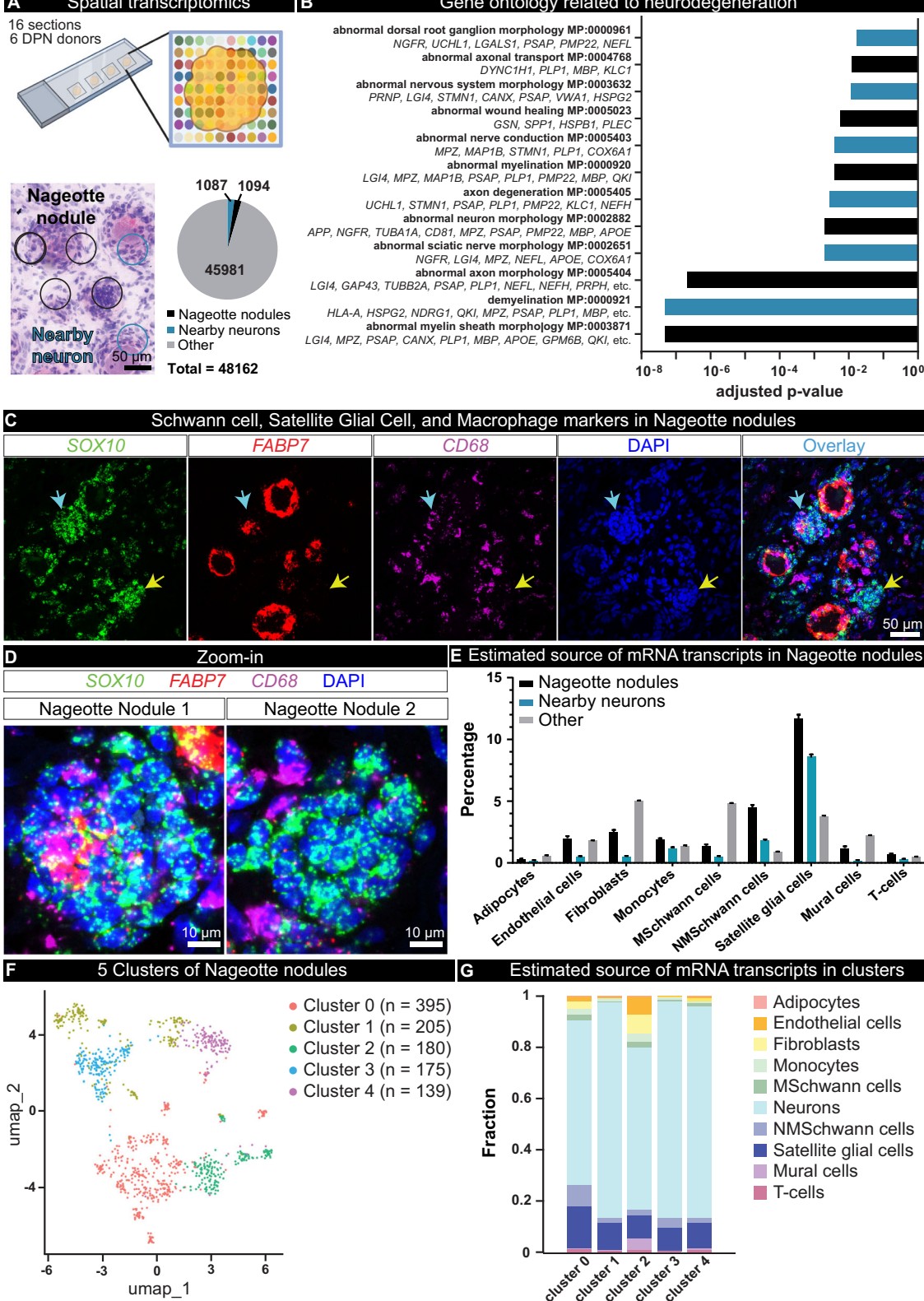

**Nageotte nodules are comprised of non-myelinating Schwann cells and satellite glia**

To elucidate the molecular profile of Nageotte nodules, we conducted spatial transcriptomics of DPN DRGs. 16 sections were processed from 6 donors; 5 donors (donors 1–4, and 6) were from the diabetic with peripheral neuropathy group, and one (donor 14) was from the diabetic taking analgesics group. Barcodes that overlapped Nageotte nodules and adjacent neurons were manually selected (Supplementary Fig. 9A, B), yielding 1094 barcodes that overlapped Nageotte nodules, and 1087 that were touching nearby neurons (Fig. 4A). Gene ontology of Nageotte nodules identified many terms related to sensory neuron degeneration (Fig. 4B) as well as other terms associated with cellular components, biological processes, and molecular functions associated with axonal sprouting and sensory neuron pathology (Supplementary

**Fig. 4 | Spatial transcriptomics of Nageotte nodules identifies non-myelinating Schwann cells and satellite glia as prominent cell types. A** VISIUM spatial transcriptomics was conducted on 16 DRG sections from 6 DPN donors. Barcodes touching Nageotte nodules (1094) and nearby neurons (1087) were selected for downstream analysis. **B)** Key gene ontology themes were related to neurodegeneration. **C** RNAscope in situ hybridization for *SOX10* (green, satellite glia and Schwann cells), *FABP7* (red, satellite glia), *CD68* (purple, macrophages), and DAPI (blue) in a DPN DRG. Confocal, 40X. Sample size: DPN *n* = 6. **D** Digitally zoomed overlay images of Nageotte nodule 1 (cyan arrow in (**C**)) and Nageotte nodule 2 (yellow arrow in (**C**)). Nageotte nodule 1 had higher content of *FABP7*+ nuclei, while Nageotte nodule 2 had little-to-no *FABP7* signal, indicating that there are differences in the composition of cell types between Nageotte nodules. **E** Deconvolution using single-nuclear sequencing datasets revealed that the potential sources of the

majority of the non-neuronal mRNA transcripts in Nageotte nodules come from Satellite glial cells and non-myelinating (NM) Schwann cells. The remaining percentage of transcripts in Nageotte nodules arise from neurons, likely axonally trafficked mRNAs. Data points are not shown on graph as they represent individual barcodes (*n* = 1041, 1064, and 39191 for nodules, nearby neurons, and other barcodes, respectively). **F** Clustering analysis of Nageotte nodule barcodes identified 5 subclusters. **G** Deconvolution reveals differences in the distribution of estimated mRNA sources between clusters. Statistical tests: **B**: Fisher's exact test with correction for multiple comparisons run in Enrichr. Adjusted *p*-values shown. **E**: Error bars = mean +/- SD. Scale bars: **C**: 50 μm. D: 10 μm. Graphic in (**A**): Created in BioRender. Shiers, S. (2025) https://BioRender.com/5u9g7kp. Source data are provided as a Source Data file.

Data 2). A full list of mRNAs detected at Nageotte nodules can be found in Supplementary Data 3. While Jean Nageotte wrote that Nageotte nodules were formed by SGCs[1], there has been no molecular characterization of these structures to confirm their cellular composition. Using cell-type marker genes identified in the spatial sequencing data, we conducted histology to validate localization of specific cell types to Nageotte nodules. Schwann cell and SGC markers like S100 and SOX10 were both detected at Nageotte nodules (Supplementary Fig. 10A), while the blood vessel marker, CD31, was not (Supplementary Fig. 10B).

In conducting these experiments, we noted that Nageotte nodules could be separated into different populations based on their expression of the SGC-specific marker, FABP7, and the SGC and Schwann cell marker, SOX10 (Fig. 4C, D). SOX10 is a transcription factor required for the differentiation of pluripotent neural crest cells into SGCs and Schwann cells[47,48], thereby, cells that are SOX10-positive, but FABP7-negative are Schwann cells, while cells expressing both are SGCs. While virtually all of the cells at Nageotte nodules were SOX10-positive, FABP7 was only detected in a subset of them (Fig. 4C, D). CD68, a macrophage marker, was also detected in a small number of peripheral cells at Nageotte nodules (Fig. 4C, D); however, CD68 along with other antigen-presenting cell markers have been reported to be expressed in SGCs in the human trigeminal ganglia[49]. Deconvolution of the Nageotte nodule barcodes using single-nuclei RNA sequencing data of human DRG cell types revealed that the highest percentage of non-neuronal mRNA transcripts at Nageotte nodules likely belonged to SGCs and non-myelinating Schwann cells, followed by fibroblasts, macrophages, endothelial cells, T-cells, and other cell types (Fig. 4E, Supplementary Data 4). Marker genes for a subset of the cell types can be visualized spatially on the H&E image in Supplementary Fig. 9C.

### Nageotte nodules express osteopontin and other ligands that could interact with nearby neurons

Because our histology findings suggested that Nageotte nodules have different cellular compositions, we examined whether we could identify unique clusters of Nageotte nodule types from spatial barcodes. We identified 5 different subclusters of Nageotte nodules (Fig. 4F) that were represented across all VISIUM slides and donors, indicating that these subclusters are not a product of batch effect nor individual differences (Supplementary Fig. 9D, E). Deconvolution analysis revealed small differences in the composition of cell types within each cluster (Fig. 4G). The most abundant of these was neuronal in each cluster. Given that there is no neuronal soma at Nageotte nodules, the neuronal signature is likely related to axonally trafficked mRNAs from nearby neurons. Some of the most highly expressed genes at Nageotte nodules were peripherin, neurofilaments, tubulins, and other cytoskeletal mRNAs that are translated locally to support axonal growth (Supplementary Data 3). The clustering approach reveals only subtle shifts in cell proportions but is consistent with histochemical observations and reveals differences in gene expression that may be important for interactions between cells in nodules and nearby

neurons that sprout into the neurodegenerative area of the Nageotte nodule. These differences may also underlie different stages of Nageotte nodule formation that cannot be discerned in post-mortem samples. Consistent with the neurodegeneration phenotype we observed for DPN DRGs, we also observed a transcriptomic signal for disease associated glia[42] characterized by high expression of *SPP1*, *APOE*, *TYROBP*, *CTSL* in Nageotte nodules (Supplementary Data 3). A subset of marker genes like *SOX10*, *SPP1*, and *CLU* can be spatially visualized on the H&E image in Supplementary Fig. 9F.

Next, we utilized the spatial transcriptomic profiles of Nageotte nodules and local surviving neurons to investigate potential ligand-receptor interactions that could be ongoing between the cells forming Nageotte nodules and neighboring neurons. We did this analysis for each of the 5 clusters since they had differences in gene expression that could influence ligand-receptor interactions. The spatial barcodes are similar in size to sensory neurons (55 μm), and we have previously been able to achieve near-single neuron transcriptomic resolution with this approach[42]; however, a limitation of the technology is that the neuronal barcodes do overlap with SGCs and other cells that ring the neurons. Thereby, some of the interactions may be representative of Nageotte ligands with receptors found on neurons and/or encircling SGCs/other cells. However, an interactome analysis can provide mechanistic insights into cellular interactions that could drive nociceptor activation, sprouting, or other cellular processes in diabetes, leading to the identification of new drug targets. First, we looked at differentially expressed ligand genes found in each cluster and examined interactions with receptors in nearby neuronal clusters. This revealed differences such as very high expression of *SPP1* in cluster 1, high expression of *CLU* in cluster 3, and a large number of neuropeptides like *CALCA*, *GAL*, *PENK*, and *ADCYAP1* also in cluster 3. A potential explanation for this finding in cluster 3 is that it is enriched in actively sprouting nociceptor axons that harbor mRNAs that might be locally translated (Fig. 5A).

Next, we assessed interactions for each cluster looking at differentially expressed receptors within nodules compared to nearby neuronal ligands. Here we found an enrichment of integrin receptor signaling in cluster 1 and observed a strong neuronal signature in cluster 3 with NTRK1-NGF signaling (Supplementary Fig. 11). Examining the most highly expressed ligands in Nageotte nodules paired with the most highly expressed receptors found in nearby neurons and then focusing on the top 50 interactions, we found striking similarities across the clusters suggesting that there are consistencies between them all among the most highly expressed genes (Supplementary Fig. 12). These included many interactions involved in neurite outgrowth or chemotaxis, such as *CLU*[50], *SPP1*[50], *CLSTN1*[51], which were found in all the clusters, and *NEGR1*, which was found in clusters 2 and 4[52,53] (Supplementary Fig. 12).

*SPP1*, the gene encoding osteopontin, was recently identified by phospho-proteomics as highly phosphorylated at multiple sites within human DPN DRGs and proposed to be involved in ER stress and extracellular matrix remodeling in DPN neurons[54]. Osteopontin mRNA

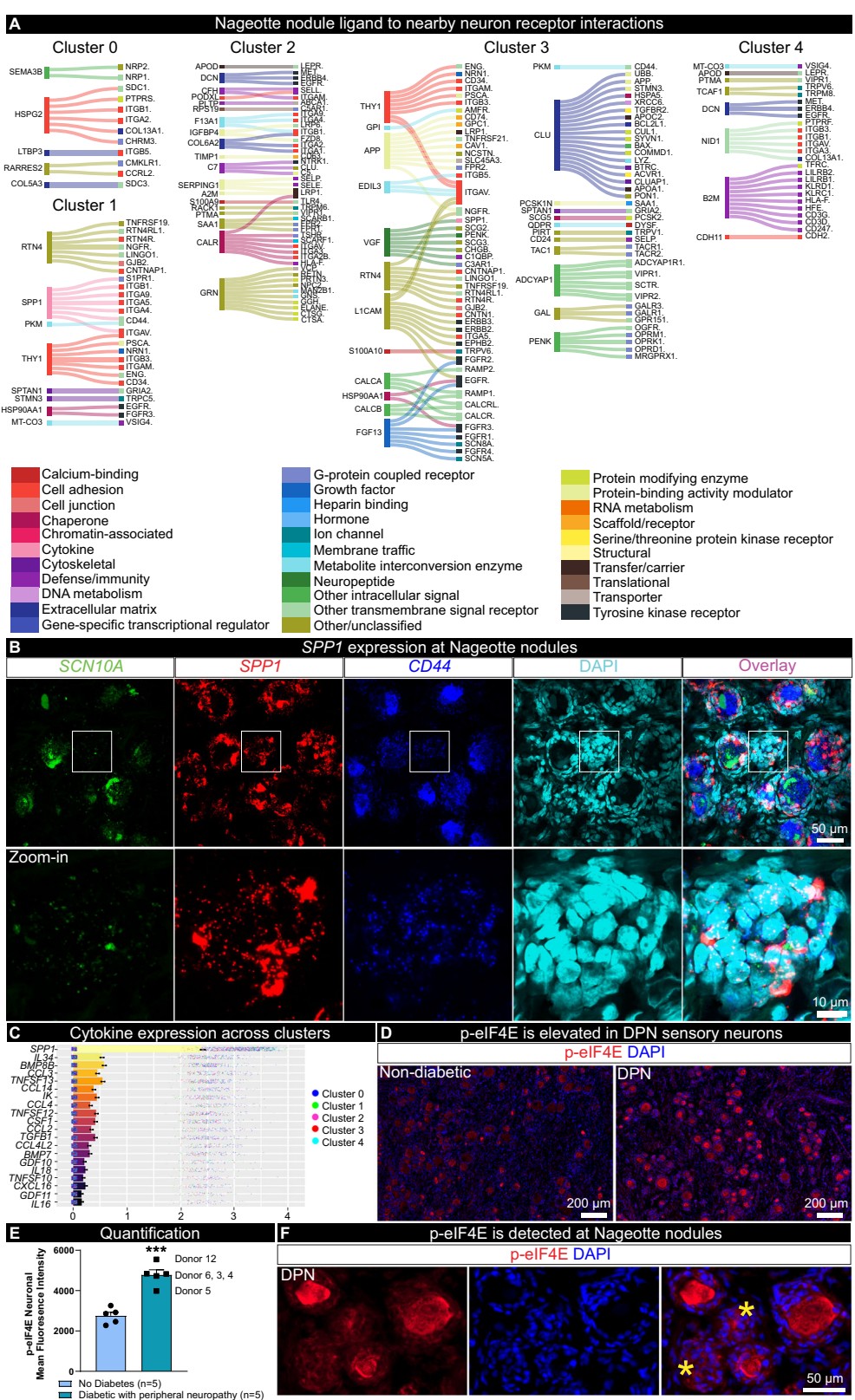

**Fig. 5** (partial). A Nageotte nodule ligand to nearby neuron receptor interactions. B SPP1 expression at Nageotte nodules. C Cytokine expression across clusters. D p-eIF4E is elevated in DPN sensory neurons. E Quantification. F p-eIF4E is detected at Nageotte nodules.

is robustly expressed in large diameter sensory neurons, SGCs, and Schwann cells in the normal, nondiabetic human DRG (Supplementary Fig. 13) and binds to many integrins and to CD44 to elicit different cellular functions such as cell migration, cell proliferation, cytokine production, and neurite outgrowth[55–59]. Our interactome data highlighted that osteopontin at Nageotte nodules may be interacting with CD44 found on nearby neurons (Fig. 5A, Supplementary Fig. 12). We

confirmed high levels of SPP1 expression in nodules with in situ hybridization (Fig. 5B), CD44 was highly expressed in most human sensory neurons (~85%) and was also found on SGCs and Schwann cells (Supplementary Fig. 13), including those forming Nageotte nodules (Fig. 5B).

While there were many cytokines expressed in Nageotte nodules, SPP1 expression was robustly detected across clusters and was the

**Fig. 5 | Ligand-receptor interactions between Nageotte nodules and nearby neurons identifies osteopontin (SPP1) and CD44. A** Differentially expressed ligands per Nageotte nodule cluster and corresponding receptors expressed in nearby neurons. **B** RNAscope in situ hybridization for *SCN10A* (green, Nav1.8), *SPP1* (red, osteopontin), *CD44* (blue), and DAPI (cyan) in a diabetic peripheral neuropathy (DPN) DRG. White outline denotes the digitally zoomed-in image of a single Nageotte nodule shown in the bottom panel. Confocal, 40X. Sample size: DPN *n* = 5. **C** Top 20 expressed cytokines in Nageotte nodules clusters, showing log10(counts per million). Data points represent individual Nageotte nodule barcodes color coded by cluster (*n* = 1094). **D** Immunohistochemistry for phosphorylated eukaryotic translation initiation factor (red, p-eIF4E) and DAPI (blue) in a non-diabetic and DPN DRG. **E** p-eIF4E was significantly elevated in the soma of sensory neurons in the DPN DRGs. Data points represent individual donors. Sample size: Non-diabetic *n* = 5, DPN *n* = 5. **F** p-eIF4E was also detected at Nageotte nodules in the DPN DRGs. Statistical tests: **E**: Unpaired two-sided *t*-test. ***p = 0.0002. **C, E**: Error bars = mean +/- SEM. Scale bars: **B**: top panel—50 µm and bottom panel—10 µm. **D**: 200 µm. **F**: 50 µm. Source data is provided as a Source Data file.

most highly expressed of this gene family in nodules (Fig. 5C). A recent study reported that SPP1 mRNA translation is dependent upon the phosphorylation of eukaryotic translation initiation factor 4E (eiF4E)[60]. Inhibition of mitogen-activated protein kinase interacting kinase (MNK), the kinase that specifically phosphorylates eIF4E, suppresses ectopic activity in human sensory neurons recovered from individuals with radicular neuropathic pain[61], and attenuates nociceptive behaviors in rodents with neuropathic pain[62–64]. We examined eIF4E phosphorylation in DPN DRGs and observed significantly increased eIF4E phosphorylation in DPN sensory neurons compared to non-diabetic controls (Fig. 5D, E). Phosphorylated eIF4E was also detected in Nageotte nodules (Fig. 5F). These findings suggest that signaling from Nageotte nodules to DRG neurons may influence the excitability state of these neurons, a common feature of neuropathic pain. Increased eIF4E signaling may also support increased osteopontin expression in DPN DRGs.

## Discussion

Our findings provide insights into the mechanisms of DPN. We find extensive neuronal degeneration in the DRGs of organ donors with DPN, with as many as 25% of neurons lost. This neurodegeneration is directly linked to the formation of Nageotte nodules that are enriched in cells with gene expression patterns that likely drive axonal sprouting and other pathological changes, such as neuronal hyperexcitability that causes pain in DPN. The current view is that axonal dieback, that is associated with pain and sensory loss in DPN can be treated with therapies that preserve axonal integrity. However, our findings reveal that neurodegeneration in DPN is extensive, suggesting that early neuroprotective strategies are almost certainly needed to protect patients from irreversible damage to DRG neurons.

Our work highlights several observations related to the nature of human sensory neurons and their response to axonal damage in diabetes. One key finding is the presence of Nageotte nodules, a structure extensively observed in the DRGs of organ donors with neuropathy but seemingly absent in rodent models of diabetes and other types of neuropathies. Notably, there are only two reports of Nageotte nodules in rat DRGs: one in which rats received multiple intravenous injections of paclitaxel[27] and another in which rats were exposed to organic mercury for 19 days[28,29]. This suggests that severe neurodegeneration may be necessary for Nageotte nodules to become readily detectable in small rodent ganglia, a process that may not be captured within the relatively short time frames (2–3 months) of typical rodent neuropathy studies. Indeed, many of the organ donors with Nageotte nodules had been living with diabetes for over 20 years, though this information was not available for all cases. Based on these findings, we hypothesize that Nageotte nodules may also form in the DRGs of diabetic or other neuropathic rodents if studied over prolonged durations. In the short term, they may be more readily studied using chemoneuropathy models, transplanted ganglia similar to Jean Nageotte's 1922 preparations[1], or using an in vitro model as Nageotte nodules appear to form in dissociated human DRG neuronal cultures.

Our second major finding is that human sensory neurons are capable of sprouting fibers from their soma and glomeruli to form pericellular nests and Nageotte nodule arborizations. It has long been observed that dissociated rodent and human sensory neurons adopt a multipolar morphology in vitro, though the significance of this phenomenon remains unclear. Since this morphology is also found in post-mortem DPN DRGs, it suggests that this phenotype may be a biological response of sensory neurons to axonal injury or cellular stress, such as the severing of the axon from the soma during DRG dissection, or from prolonged hyperglycemia as is likely the case in DPN. Regardless of the cause, the loss of a pseudounipolar shape likely alters the functional properties of these neurons and raises the compelling question of whether or not the sprouts originating from the soma and glomerulus are capable of ectopic activity, or if that activity can even be propagated through the large neuronal soma, stem axon, and central axonal branch to the spinal cord. Addressing this question will be challenging because traditional patch clamp electrophysiology on DRG neurons is done on neurons without axons, and recordings of these intact structures will likely require development of new techniques to understand the electrophysiology of Nageotte nodules.

A third major finding is the discovery of a potential ligand to receptor signaling network that emerges between SGCs and Schwann cells within the Nageotte nodule and sprouting axons that infiltrate the core of the nodule. In our view, there are several questions that should be addressed to better understand the role of these interactions in DPN and potentially other neuropathies. First, experiments aimed at understanding the influence of factors like osteopontin and clusterin[50] on the axonal sprouting of adult human DRG neurons would give insight into whether these factors drive the axonal sprouting that occurs in Nageotte nodules in humans in vivo. Second, these and other factors can be studied in electrophysiology or signaling experiments to understand whether these ligand-receptor interactions might cause generation of action potentials in DRG nociceptors to cause pain in DPN. Finally, it is also possible that specific ligand-receptor interactions cause programmed cell death in human DRG neurons. This can also be studied using human DRG neuronal cultures based on the data generated in this work. We acknowledge that these questions remain unanswered and therefore also represent important limitations of the work described here.

Overall, our work updates the pioneering work of Jean Nageotte, showing that the structures he described in the 1920s are a key pathology in the DRG in DPN. Applying the power of modern microscopy and RNA-sequencing technologies to Nageotte's nodules gives insight into the richness of the underlying biology of these small, yet extraordinarily complex structures. We propose that the work of Jean Nageotte over 100 years ago may be key to understanding and treating the most common form of neuropathic pain on Earth, DPN.

## Methods

### Human tissue procurement and ethics

All human tissue procurement procedures and ethical regulations were approved by the Institutional Review Board at the University of Texas at Dallas under protocol Legacy-MR-15-237. DRGs were procured from organ donors through a collaboration with the Southwest Transplant Alliance, an organ procurement organization (OPO) in Texas, or purchased from AnaBios (2 DRGs). The procurement network of AnaBios Corporation includes only US-based OPOs and hospitals. The Southwest Transplant Alliance, Anabios, and its OPO partners obtain informed consent for research tissue donation from

first-person consent (driver's license or legally binding document) or from the donor's legal next of kin. Policies for donor screening and consent are those established by the United Network for Organ Sharing (UNOS). OPOs follow the standards and procedures established by the US Centers for Disease Control (CDC) and are inspected biannually by the Department of Health and Human Services (DHHS). The distribution of donor medical information is in compliance with HIPAA regulations to protect donor privacy. All transfers of donor tissue to AnaBios are fully traceable and periodically reviewed by US Federal authorities.

DRGs and other nervous tissues (sympathetic chain ganglia) were recovered using a ventral approach as previously described[38,65]. Upon removal from the body, DRGs were prepared three different ways. One, DRGs used for histology and sequencing work were frozen in pulverized dry ice, transferred into prechilled epitubes, and stored in a −80 °C freezer. Two, DRGs used for neuronal dissociation were placed into freshly prepared artificial cerebral spinal fluid (aCSF) over ice. Three, DRGs used for transmission electron microscopy were placed into 4% paraformaldehyde for 48 h at 4 °C and then transferred to 1X Phosphate Buffered Saline (PBS) and shipped on ice to the University of Adelaide in Australia. A detailed protocol of the procurement process, including recipes for aCSF, can be found on protocols.io[38]. Two DRGs were purchased from Anabios, a commercial vendor that also procures nervous tissues from organ donors.

De-identified donor information was provided by the Southwest Transplant Alliance and Anabios and includes medical details from the donor's family members and hospital records. Donor information including medical history and DRG level details (majority are lumbar DRGs) is provided in Supplementary Data 1. DRGs from 92 organ donors (39 female, 53 male) with mixed race/ethnicity (62 white, 15 Hispanic, 13 black, 1 Alaskan Indian, and 1 not reported) were used for experiments. Sex at birth of each organ donor was provided by the Southwest Transplant Alliance or Anabios who obtained this information from hospital records and the donor's family. Sex was not considered in the study design as donor groupings (the DPN group, for instance) would not be properly powered for a sex-specific analysis. The frozen DRGs were gradually embedded in OCT in a cryomold by adding small volumes of OCT over dry ice to avoid thawing. Tissues were sectioned on a cryostat and utilized for histology and spatial transcriptomics. After sectioning, the remaining tissue blocks were wrapped in tin foil and then returned to the −80 °C freezer for future use.

## Hematoxylin and eosin staining, imaging, and analysis

A single DRG from 90 organ donors was sectioned at 20μm onto SuperFrost Plus charged slides (Fisher Scientific; Cat 12-550-15). The donors were randomly selected from our tissue bank as part of routine tissue morphology checking as part of our quality control process. Lumbar DRGs were preferentially selected when available in our tissue bank, but in some cases, lower thoracic DRGs were used. The DRG levels that were assessed are indicated in Supplementary Data 1. Sections were only briefly thawed in order to adhere to the slide but were immediately returned to the −20 °C cryostat chamber until completion of sectioning. The slides were removed from the cryostat and immediately immersed in 10% formalin (Fisher Scientific; Cat 23-245684) for 15 min. The tissues were then sequentially dehydrated in 50% ethanol (5 min; Fisher Scientific; Cat 04-355-223), 70% ethanol (5 min), and two times in 100% ethanol (5 min) at room temperature. The slides were air dried briefly, and then each section was covered with isopropanol (Sigma; L9516) and incubated for 1 min at room temperature. The excess isopropanol was removed, and the slides were allowed to air dry again briefly (<5 min). Hematoxylin (Sigma; MHS16) was applied to each tissue section until covered and incubated for 7 min at room temperature. The excess Hematoxylin was removed by tapping, and the slides were immersed 30 times in ultrapure water (dipping into the

water). Bluing Buffer (Agilent; CS70230-2) was applied to each tissue section until covered and incubated for 2 min at room temperature. The excess Bluing Buffer was removed by tapping, and the slides were immersed 5X in ultrapure water (dipping into the water). Eosin mix (1:10 of 0.45 M Tris Acetic Acid Buffer to Eosin (Sigma; HT110216)) was applied to each tissue section until covered and incubated for 1 min at room temperature. The excess Eosin mix was removed by tapping, and the slides were immersed 15X in ultrapure water (dipping into the water). The excess water was removed from the slide using a tapping motion and Kim wipe. The slides were allowed to completely air dry before being coverslipped with Prolong Gold Antifade (Fisher Scientific; Cat P36930).

DRG sections were mosaically imaged at 10X using default brightfield settings on an Olympus vs120 slide scanner. The raw images were opened in CellSens (Olympus; v1.18) and qualitatively scored for the presence of Nageotte nodules throughout the entire DRG section. A qualitative scoring system (ranked from high to very low) was developed by comparing sections to one another and noting the extremes: DRG sections that had an abundance of Nageotte nodules (high) versus those with very little-to-none (very low). The person analyzing was blinded to the donor's demographics and medical history. Once each section was scored, the medical information from each donor was probed and grouped into categories based on diagnoses of diabetes or other pain conditions.

Diabetes diagnosis is known for each donor as insulin/blood sugar is monitored while the donors are on life support. Since the donor's medical history is a summary of hospital records and information from the next of kin provided by the organ procurement organization, information about neuropathy or pain is sometimes not reported. As such, we used keywords to group donors and make inferences about DPN. These keywords are bolded in Supplementary Data 1.

We grouped the donors into 5 categories: diabetics with peripheral neuropathy (medical history statements of having neuropathy, nerve pain, and/or amputation), diabetics taking analgesics (analgesic(s) usage indicated in medical history but use for neuropathy is not specified, no obvious signs of drug abuse/addiction), diabetics not taking analgesics (no report of taking analgesics for pain/neuropathy, drug abuse/addiction included), non-diabetics (no DRG-affiliated pain condition, and no diabetes), non-diabetics with other pain conditions (fibromyalgia, arthritis, neuropathy, back pain as indicated in medical history). The donors included in each category are indicated in Supplementary Data 1.

## Pre-mounted section immunofluorescence staining, imaging, and analysis

3-4 20μm tissue sections (technical replicates) were acquired from each DRG (sample sizes indicated in figure captions) and placed onto SuperFrost Plus charged slides (Fisher Scientific; Cat 12-550-15). Slides were removed from the cryostat and immediately transferred to cold 10% formalin (pH 7.4) for 15 min. The tissues were then dehydrated in 50% ethanol (5 min), 70% ethanol (5 min), 100% ethanol (5 min), 100% ethanol (5 min) at room temperature. The slides were air dried briefly, and then boundaries were drawn around each section using a hydrophobic pen (ImmEdge PAP pen, Vector Labs). When hydrophobic boundaries had dried, the slides were submerged in blocking buffer (10% Normal Goat Serum, 0.3% Triton-X 100 in 1X PBS) for 1 h at room temperature. Slides were then rinsed in 1X PBS, placed in a light-protected humidity-controlled tray, and incubated in primary antibody diluted in blocking buffer overnight at 4 °C. A list of all primary and secondary antibodies is shown in Supplementary Data 5. The next day, slides were washed in 1X PBS and then incubated in their respective secondary antibody (1:2000) with DAPI (1:5000; Cayman Chemical; Cat # 14285) diluted in blocking buffer for 1 h at room temperature. The sections were washed in 1X PBS and then covered with True Black (diluted at 1:20 in 70% Ethanol; Biotium; 23014), a

blocker of lipofuscin, for 1 min. Sections were then rinsed vigorously with ultrapure water and then washed in 1X PBS. The slides were then air dried and coverslipped with Prolong Gold Antifade reagent. A negative control consisting of 1 section from each DRG was processed in every immunohistochemistry experiment and was exposed to all of the same reagents except for primary antibody. In the negative controls, unremarkable autofluorescence was observed in the green channel but otherwise, all channels were virtually void of any signal. Representative images of negative controls can be seen in Supplementary Fig. 14.

DRG sections were imaged on a vs120 slide scanner (Evident Scientific) or an FV3000 or FV4000 confocal microscope (Evident Scientific) at 10X, 20X, 40X, 60X, or 100X magnification as indicated in the figure captions. The acquisition parameters were set based on guidelines for the vs120, FV3000, and FV4000 provided by Evident Scientific. The raw image files were brightened and contrasted in Olympus CellSens software (v1.18) for display purposes. For quantification and DPN vs non-diabetic comparison experiments, all acquisition and brightness/contrast adjustment parameters were kept the same in order to make direct comparisons between samples.

For the Nageotte nodule quantification experiment (GFAP + Peripherin IHC), all neurons with a visible cytoplasm (peripherin+) were counted, and all of the Nageotte nodules were counted (cluster of DAPI+ nuclei that was GFAP+ and negative for a peripherin+ neuronal soma) in Olympus CellSens (v1.18). The percentage of neurons with a Nageotte nodule morphology was calculated by dividing the Nageotte nodule counts by the total neuronal population (sum of Nageotte nodules and neurons) and multiplying by 100. Three 20X mosaic sections (vs120) were analyzed per donor, and then the final percentages from each section were averaged for each donor. A random names script was used to rename all image files. The script also provides a translation file with the original file names and its new file ID to be referenced after analysis. This script is publicly available and was published Nov 3, 2016 on how-to-geeks.com by Jason Faulkner.

For the neuronal size measurement experiment (GFAP + Peripherin IHC), we measured the diameter of neurons that had a peripherin+ soma, were ringed by GFAP+ cells, and had a visible nucleus using the polyline tool in Olympus CellSens (v1.18). An analyzer who was blinded to the donor's medical history information and their associated groupings performed the analysis. The number of Nageotte nodules was also counted in each image. Neurons were grouped as small (<65 μm), medium (65–75 μm), or large (75 μm+) based on the size profile of C-nociceptors, Aδ neurons, and Aβ neurons as we have previously reported on in human DRG[42]. The percentage of small, medium, and large-sized neurons was calculated by dividing the number of neurons in each group by the total neuronal population (sum of neurons with visible nuclei + Nageotte nodules). One image (2 mm × 1.5 mm) was analyzed per section (each containing -100 neurons), and three sections were analyzed per donor. The final percentages from each section were averaged for each donor.

For peripherin fiber density analysis, a single 10X confocal image (FV3000) was acquired for each section of DRG (three sections total from each DRG, 3 images/donor). A single negative control DRG section from each donor (exposed to all reagents except for primary antibody) was imaged with the same settings. The neuron-rich area of the DRG was manually outlined in Olympus CellSens (v1.18), and its area was provided by the software. The peripherin signal within the neuron-rich area was autodetected using the Count and Measure feature in Olympus CellSens (v1.18). The peripherin signal within the soma of the neurons was then manually removed in the software. The remaining area of peripherin signal (axonal only) was provided by the CellSens software (v1.18) and divided by the neuron-rich area for each section, and then averaged across all sections for each DRG.

For the p-eIF4E experiment analysis, a mosaically tiled 20X image (Evident Scientific, vs120) was acquired for each section of DRG (three sections total from each DRG, 3 images/donor). A single negative control DRG section from each donor (exposed to all reagents except for primary antibody) was imaged with the same settings. The cell body of all neurons within the field of view were manually outlined using the Closed Polygon tool in Olympus CellSens (v1.18), and the software output the mean fluorescence intensity of the p-eIF4E signal within the ROI. This was performed on both the experimental and negative control sections. For each donor, the final p-eIF4E mean fluorescence intensity value was corrected by subtracting the mean fluorescence intensity value of the negative control.

## Filament tracing and free-floating immunofluorescence staining

A DPN donor with the highest Nageotte score (Donor #6) was selected for filament tracing in order to grant the highest likelihood of tracing Nageotte nodule fibers to their point of origin. For free-floating immunofluorescence staining, 50 μm DRG sections were acquired on a cryostat and then immediately submerged in 10% formalin (pH 7.4) in a 24-well plate. The sections were fixed for 15 min, and then washed in an adjacent well containing 1X PBS. The sections were then transferred to a well containing blocking solution (10% Normal Goat Serum, 0.3% Triton-X 100 in 1X PBS) for 1 h at room temperature before being transferred to primary antibody (peripherin, Supplementary Data 5) diluted in blocking solution over night at 4 °C.

The next day, the sections were washed in 1X PBS, and then placed into a well containing secondary antibody (1:2000, Supplementary Data 5) with DAPI (1:5000) diluted in blocking buffer for 1 h at room temperature while being shielded from light. The sections were then washed in 1X PBS, mounted onto slides, and treated with True Black (diluted at 1:20 in 70% Ethanol; Biotium; 23014) for 1 min. Sections were then rinsed vigorously with ultrapure water and then washed in 1X PBS. The slides were then air dried and coverslipped with Prolong Gold Antifade reagent.

60X z-stack images with optimal z-slices (0.3 μm) of the entire z plane were acquired on an FV4000 confocal microscope (Evident Scientific). The images were loaded into Imaris (v10), converted to IMS files, and then the peripherin signal was traced using the semi-automatic filament tracing tool. Starting points were manually selected at fibers within the Nageotte nodules and at a neighboring neuron's soma. The fibers were traced using the semi-automatic filament path finding tool which path finds continuous peripherin signal through the z-plane. Only fibers that originated from the Nageotte nodule arborizations or the neuronal soma and had paths that were continuously autodetected through the z-plane were traced. Other axons in the image that were not traced were either discontinuous or did not originate from the designated starting points.

## RNAscope in situ hybridization staining, imaging, and analysis

DRGs were sectioned at 20μm onto SuperFrost Plus charged slides (Fisher Scientific; Cat 12-550-15). Sections were only briefly thawed in order to adhere to the slide but were immediately returned to the -20 °C cryostat chamber until completion of sectioning. The slides were removed from the cryostat and immediately immersed in cold (4 °C) 10% formalin (Fisher Scientific; Cat 23-245684) for 15 min. The tissues were then sequentially dehydrated in 50% ethanol (5 min; Fisher Scientific; Cat 04-355-223), 70% ethanol (5 min), and two times in 100% ethanol (5 min) at room temperature. The slides were air dried briefly, and then boundaries were drawn around each section using a hydrophobic pen (ImmEdge PAP pen; Vector Labs). Once the hydrophobic boundaries had dried, the slides were immediately processed for RNAscope in situ hybridization.

RNAscope in situ hybridization multiplex version 2 (Advanced Cell Diagnostics; Cat 323100) was conducted on human DRGs using the fresh frozen protocol as described by ACD (acdbio; manual #

323100-USM with rev date: 02272019). Hydrogen Peroxide (ACD; Cat 322381) was applied to each section until fully covered and incubated for 10 min at room temperature. The slides were then washed in distilled water and then were incubated one at a time in Protease III reagent (ACD; Cat 322381) for 10 s at room temperature. The protease incubation time was optimized as recommended by ACD for the tissue and specific lot of Protease reagent. Slides were washed briefly in 1X phosphate buffered saline (PBS, pH 7.4) at room temperature. Each slide was then placed in a prewarmed humidity control tray (ACD; Cat 321710) containing dampened filter paper (ThermoFisher Scientific; Cat 84784) and a mixture of Channel 1, Channel 2, and Channel 3 probes (50:1:1 dilution as directed by ACD due to stock concentrations) was pipetted onto each section until fully covered. This was performed one slide at a time to avoid liquid evaporation and section drying. The humidity control tray containing the slides was placed in a HybEZ oven (ACD; Cat 321710) for 2 h at 40 °C. A table of all probes used is shown in Supplementary Data 5. Following probe incubation, the slides were washed two times in 1X RNAscope wash buffer (ACD; Cat 310091) and then placed in 5X SSC buffer (Sigma; Cat S6639) over night at room temperature.

The following morning, the slides were washed two times in 1X RNAscope wash buffer (ACD; Cat 310091) and placed in the 40 °C oven for 30 min after submersion in AMP-1 reagent. Washes and amplification were repeated using AMP-2 and AMP-3 reagents with a 30-minute and 15-minute incubation period, respectively. HRP-C1 reagent was applied to all sections and then incubated in the oven at 40 °C for 15 min. The slides were then washed in 1X RNAscope wash buffer (ACD; Cat 310091). TSA Plus Akoya Dyes in Fluorescin, Cyanine-3, and Cyanine-5 (Akoya; NEL741001KT, NEL744001KT, NEL745001KT) were prepared at 1:1000 in TSA buffer (ACD; Cat 322809). The Akoya dye assigned to Channel 1 probe was applied to each section until fully covered and incubated for 30 min in the 40 °C oven. The slides were washed and then covered in HRP blocker (ACD; Cat 323110) for 15 min at 40 °C. The slides were washed again, and then the same steps were repeated using HRP-C2 and HRP-C3 reagents with their assigned Akoya dye. DAPI (ACD; Cat 323110) was applied to each section for 1 min at room temperature and then washed in 1X PBS (pH 7.4) before being washed, air dried, and cover-slipped (Globe Scientific; Cat 1415-15) with Prolong Gold Antifade mounting medium (Fisher Scientific; Cat P36930).

A positive and negative control was run on a single section from each DRG for every RNAscope experiment. The positive control probe cocktail (Supplementary Data 5) contains probes for high, medium, and low-expressing mRNAs that are present in all cells (ubiquitin C > Peptidyl-prolyl cis-trans isomerase B > DNA-directed RNA polymerase II subunit RPB1) and allows us to gauge tissue quality and experimental conditions. All tissues showed robust signal for all 3 positive control probes. A negative control probe against the bacterial DapB gene (Supplementary Data 5) was used to check for lipofuscin and background label.

DRG sections were imaged on an FV3000 or FV4000 confocal microscope (Evident Scientific) at 10X, 20X, 40X, or 60X magnification as indicated in the figure captions. The acquisition parameters were set based on guidelines for the FV3000 and FV4000 provided by Evident Scientific. The raw image files were analyzed in CellSens (Olympus; v1.18). For quantification analyses, three 20X confocal images were analyzed per section, and three sections were imaged per donor. Target positive neurons were considered to be neurons with >5 puncta for each channel. The True black lipofuscin quencher (used in Immunofluorescence) is not compatible with RNAscope. Large globular structures and/or signal that auto-fluoresced in all channels (particularly brightest in 488 and 555 wavelengths) were considered to be background lipofuscin and were not analyzed. Aside from adjusting brightness/contrast, we performed no digital image processing to subtract background.

## Transmission electron microscopy tissue preparation, staining, and imaging

Upon receipt from UTDallas, the DRGs were cut into smaller pieces of about 1 mm$^3$ in size, fixed in 2.5% glutaraldehyde in phosphate buffer, pH 7.4 at 4 °C for 24 h, washed in PBS, and transferred into 2% aqueous osmium tetroxide solution for 1 h. The samples were then dehydrated in a graded series of ethanol and embedded in TAAB epon araldite embedding medium at 60 °C for 48 h. Ultrathin sections of 70–90 nm thickness were cut using a ultramicrotome (Leica), stained with 4% uranyl acetate and Reynolds lead citrate for 8 min, and examined using an electron microscope (FEI Tecnai 120 kV Spirit). Images were captured using an AMT Camera with AMT_V7.0.1 software.

## Dissociated neuronal cultures and Immunocytochemistry

After procurement, DRGs (donors 91 and 92, Supplementary Data 1) in aCSF were transported to the lab over ice (~30 min). The DRGs were trimmed of excess connective tissue, fat, and nerve roots to expose the DRG bulb. The DRG bulb was then cut into 3 mm sections and placed in 5 mL of pre-warmed digestion enzyme containing 1 mg/mL of Stemzyme I (Worthington Biochemical, LS004106), 0.1 mg/mL of DNAse I (Worthington Biochemical, LS002139), and 10 ng/mL of recombinant human β-NGF (R&D Systems, 256-GF) in HBSS without calcium and magnesium (Thermo Scientific, 14170-112). The tubes were placed in a 37 °C shaking water bath until the DRG sections dissociated (4–10 h). The solution was then filtered through a 100 μm mesh strainer. The resultant cell suspension was then gently added to a 15 mL tube containing 3 mL of 10% Bovine Serum Albumin (Biopharm, 71-040) in HBSS. The tubes were then centrifuged at 900 g for 5 min at room temperature. The supernatant was aspirated, and the pellet was resuspended in prewarmed BrainPhys® media (Stemcell technologies, 05790) containing 1% penicillin/streptomycin (Thermo Fisher Scientific, 15070063), 1% Glutamax (Thermo Scientific, 35050061), 2% NeuroCult SM1 (Stemcell technologies, 05711), 2% HyClone™ Fetal Bovine Serum (Thermo Fisher Scientific, SH3008803IR), 1% N-2 (Thermo Scientific, 17502048), 0.1% 5-Fluoro-2'-deoxyuridine (FRDU, Sigma-Aldrich, F0503), and 10 ng/ml of β-NGF. Cells were plated in a 24-well plate containing 12 mm coverslips which were pre-coated with 0.1 mg/mL of poly-D-lysine at a seeding density of 100 neurons per well. Cells were incubated at 37 °C and 5% CO$_2$ for 3 h to allow for adherence. Following neuron adherence, wells were flooded with 1 mL of prewarmed media, and half media changes were performed every other day.

On DIV 3 (donor 91) and DIV 5 (donor 92), cells were washed with 1X PBS and fixed with 10% formalin for 10 min at room temperature. Cells were then washed 3 times with 1X PBS and blocked with 10% normal goat serum in PBS for 1 h at room temperature. Cells were then permeabilized with 10% Normal Goat Serum and 0.3% Triton X in PBS for 30 min at room temperature. To label neurons, cells were incubated with peripherin (1:1000, Supplementary Data 5) diluted in blocking buffer overnight at 4 °C. The next day, cells were washed 3 times with 1X PBS and incubated with a goat anti-chicken 488 secondary antibody (1:2000) and DAPI (1:5000, Cayman Chemical, 14285) diluted in blocking buffer for 1 h at room temperature. Cells were washed with 1X PBS and then covered with True Black (diluted at 1:20 in 70% Ethanol; Biotium; 23014), a blocker of lipofuscin, for 1 min. The cells were washed again in 1X PBS. Coverslips were lifted out of the 24-well plate and mounted onto glass slides with Prolong Gold Antifade reagent (Fisher Scientific, P36930). A negative control coverslip was processed similarly in each experiment but was not exposed to the primary antibody. All images were taken on an Olympus FV3000 confocal microscope at the University of Texas at Dallas.

## Spatial transcriptomics

VISIUM tissue optimization and spatial gene expression (v1) protocols were followed exactly as described by 10x Genomics (https://www.

10xgenomics.com/) using Haematoxylin and Eosin as the counterstain. Optimal permeabilization time was obtained at 12 min incubation with permeabilization enzyme[42]. Imaging was conducted on an Olympus vs120 slide scanner. mRNA library preparation and sequencing (Illumina Novaseq, NextSeq 500, NextSeq 2000) were done at the Genome Center in the University of Texas at Dallas Research Core Facilities. DRGs from 6 DPN donors were used and processed in three separate VISIUM experiments (donors 1–4, 6, 14). Sections from each donor were 200 µm apart so as to not sample the same neurons or Nageotte nodules across sections. 16 sections were processed for VISIUM: 2 from donor 1, 4 from donor 2, 3 from donor 3, 3 from donor 4, 2 from donor 6, and 2 from donor 14. The number of sections used was based on tissue availability and the number of unoccupied etched frames that were available on each slide in each experiment. The demographics and medical information for each donor are provided in Supplementary Data 1. Raw sequencing files were processed with the 10X Genomics SpaceRanger pipeline (versions 1.1.0, 1.3.0, and 2.0.0) to generate count matrices of gene expression per VISIUM barcode for 16 DRG sections from 6 donors. VISIUM sections were examined in Loupe Browser (10X Genomics), and barcodes that overlapped Nageotte nodules and nearby neurons were manually selected. Barcodes that overlapped individual or multiple Nageotte nodules were annotated as "Nageotte Nodules." Barcodes that overlapped Nageotte nodules and neurons were annotated as "Nageotte also touching neuron" and were not included in the analysis. Barcodes touching the soma of a neuron that was near a Nageotte nodule (within 1 barcode) were annotated as "nearby neurons." All barcodes that did not fit into these categories were considered to be "Other" barcodes. The annotated barcodes with instructions on how to import them into the downloadable Loupe Browser files can be found in Supplementary Data 6. Overall, 1094 Nageotte nodule barcodes and 1087 nearby neuronal barcodes were selected. Spatial sequencing metrics can be found in Supplementary Data 7.

### Spatial RNA-seq analysis

Gene expression analysis of the barcodes was done with R (version 4.3.3) and consisted of quantification of gene expression in Nageotte Nodules, Enrichr analysis of top genes, spatial deconvolution, clustering of Nageotte nodule barcodes, and interactome analysis.

**Analysis of gene expression and top genes in Nageotte nodules.** The raw counts from all nodule barcodes were pseudo-bulked and normalized to library size to generate counts per million (CPM) values for each gene in the Nageotte nodule barcodes. As the 10X VISIUM assay library preparation does not have a gene length bias, the CPM values were deemed sufficient to allow comparisons between genes. A filtered table was generated that included only protein-coding genes (indicated by Ensembl gene biotype) and excluded genes from the mitochondrial chromosome and those that code for ribosomal proteins. After filtering, the gene expression values were re-normalized to sum one million.

A recently published dataset from our group of single-nuclei RNA-sequencing from human DRGs[66] was processed with the Seurat[67] (version 5.0.3) FindAllMarkers() function using the Wilcoxon rank sum test and default parameters to determine a list of neuronally enriched genes using the following criteria: 1) a difference of at least 10% when subtracting the percentage of non-neuronal cells that express the gene from the percentage of neurons that express the gene, 2) a log2 fold-change of at least 1 in gene expression between non-neurons and neurons, 3) adj. p-value < 0.05 in neurons and >0.05 in every other cell type. This list was intersected with the filtered Nageotte nodule gene expression table to generate a table with 491 neuronally enriched genes and a table with 18714 non-enriched genes.

A curated list of 139 cytokine genes, largely consisting of those genes classified as cytokines in the PANTHER database[68], was

intersected with the gene expression table and the expression of the top 20 genes across all Nageotte barcodes is shown in Fig. 5E. All data points with a value of 0 in this plot correspond to barcodes with no expression of the gene and were given this value to allow their visualization after log10 transformation of the counts per million value. All gene expression tables can be found in Supplementary Data 3.

**Enrichr analysis.** The top 300 genes in the filtered gene expression table were analyzed with the Enrichr[69] website (https://maayanlab.cloud/Enrichr/) to determine ontology terms enriched in the gene set in the following databases: 1) MGI Mammalian Phenotype Level 4 2021, 2) GO Molecular Function 2023, 3) GO Biological Process 2023, 4) GO Cellular Compartment 2023. Enrichr uses 4 statistical algorithms to determine significance including a Fisher's exact test with a correction for multiple comparisons. The adjusted p-value, which reflects correction for multiple comparisons, was used to identify processes that are associated with highly expressed genes in Nageotte nodules, and the full output can be found in Supplementary Data 2.

**Spatial deconvolution.** All barcodes from the 16 sections with greater than 200 unique genes were processed with SONAR[70] (version 1) to predict cell type proportions using a signature matrix of marker genes per cell type. The signature matrix was generated by subsetting the raw count matrix of a single-nuclei dataset of human DRG cells to include only those genes that were highly enriched in a cell type (log-fold change > 1.0, expression in > 50% of cells, and adj. p-value < $10^{-20}$). Deconvolution results were used to estimate the contribution of different cell types to the transcriptomes of barcodes with Nageotte nodules, adjacent neurons, and all other barcodes (barcodes that overlapped both nodules and neurons were excluded from all 3 categories). They were also used to predict cell type composition of the different clusters of Nageotte nodules.

**Clustering of Nageotte nodule barcodes.** The SCT pipeline from Seurat (version 5.0.3), followed by Harmony integration and FindNeighbors()/FindClusters(), was used to perform unsupervised clustering of the Nageotte nodule barcodes. A UMAP plot of the clustering with cells labeled by donor was used to confirm that the Nageotte barcodes did not exhibit donor-specific groupings. The cells in each cluster were pseudo bulked to obtain expression in CPM for all genes per cluster, and FindMarkers() was used to determine enriched genes per cluster.

**Interactome analysis.** To explore potential signaling mechanisms between Nageotte nodules and neighboring neurons, reads from neuronal barcodes adjacent to the nodules were pseudo bulked and normalized to CPM. A curated ligand-receptor database and interactome platform (https://sensoryomics.shinyapps.io/Interactome/;[71]) were used to hypothesize interactions with ligands from each Nageotte nodule cluster and receptors on the neurons adjacent to nodules. In the first interactome analysis, to determine the top potential interactions per cluster, the interactions were ranked based on the sum of the ligand and receptor CPM expression values. In the second analysis, to determine interactions that may be more prominent for each cluster, only ligand genes that were differentially expressed (adj. p-value < 0.05) in each cluster were used. Lastly, we also observed potential interactions with ligands from the surrounding neurons and receptors on the nodules (using only those receptor genes that were enriched in a cluster with adj. p-value < 0.05). Ligand and receptor genes were also labeled with the protein class of their gene product using the PANTHER database[68].

### Statistics and reproducibility

Graphs were generated using GraphPad Prism version 8.4.3 (GraphPad Software, Inc. San Diego, CA, USA). Statistical analyses (either unpaired

two-sided t-tests or one-way or two-way ANOVA) were run in GraphPad Prism as indicated in the figure captions. Sample size is also indicated on the graphs and/or figure captions. Graphical figures were made in Biorender under an academic license as indicated in the figure caption. Bar graphs were generated with the R ggplot2 package to visualize expression of genes in Nageotte nodule barcodes. For figures that display fluorescent representative images, three technical replicates (tissue sections) were generated per biological replicate (each organ donor), and multiple biological replicates were tested in each experiment as indicated in the main text and figure captions. All biological and technical replicates gave similar staining patterns unless otherwise noted in the main text and figure caption. If fluorescence image variability was seen across donors or if a qualitative analysis was used, then additional images are provided as Supplementary Figs. which demonstrates the representative signal across donors, such as in the Nav1.7, TH, and TrpV1 experiments.

## Data availability
The unprocessed sequencing data generated in this study are available on the dbGap database under the accession code phs001158. The processed sequencing data are publicly available on GEO under the accession code GSE295206. Source data are provided with this paper.

## Code availability
All code is deposited on GitHub at https://github.com/utdal/Nageotte-Nodule-Analysis.

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

## Acknowledgements

The authors thank the organ donors and their families for their gift, members of the Southwest Transplant Alliance for supporting the tissue recovery work, and members of the Price lab for useful discussions. This research was supported by the National Institute of Neurological Disorders and Stroke of the National Institutes of Health through the PRECISION Human Pain Network (RRID:SCR_025458), part of the NIH HEAL Initiative (https://heal.nih.gov/) under award number U19NS130608 to TJP. This work was also funded by NIH grant R01NS111929 to TJP. The content is solely the responsibility of the authors and does not necessarily represent the official views of the National Institutes of Health.

## Author contributions

S.S. and T.J.P. wrote the manuscript, S.S. did the histology experiments and analysis, DTF/IS/SS conducted spatial transcriptomics, K.M. and A.W. analyzed the spatial sequencing data, R.H. conducted electron microscopy experiments, N.E. assisted in image analysis, J.L. conducted the cell culture experiments, G.D. provided scientific/experimental input, A.C./G.F./P.H./E.V. coordinated human tissue recoveries, performed dissections of DRGs from organ donors, assisted in DRG tissue processing for data acquisition, and provided scientific input.

## Competing interests

T.J.P. is a co-founder of 4E Therapeutics. The authors declare no other competing interests related to this work.
