## [Transparent Peer Review file · Nature Communications]

Nageotte nodules in human dorsal root ganglia reveal neurodegeneration in diabetic peripheral neuropathy

Corresponding Author: Dr Theodore Price

Version 0:

Reviewer comments:

Reviewer #1

(Remarks to the Author)

In this study, the authors used elegant spatial sequencing and histology approaches to uncover the cellular and molecular nature of Nageotte nodules from DRG tissue of organ donors with DPN. They found that these structures primarily consist of satellite glial cells and Schwann cells and that they may contribute to painful neuropathy pathogenesis by regulating axonal sprouting and hyperexcitability. The paper presents a novel and timely contribution to the DPN field, addressing pain and sensory loss, which are currently understudied. However, I have a few comments that I believe should be addressed to enhance the clarity and rigor of the work:

1. Currently, the introduction treats Nageotte nodules and diabetic neuropathy pathogenesis as two separate topics, which may make it difficult for readers to appreciate their interrelationship, which is a main theme of the study. Consequently, I recommend better integrating the discussion of how elucidating the role of Nageottes nodules could enhance our understanding of painful diabetic neuropathy. This connection will improve introduction clarity and provide a stronger context for the readers.
2. The quality of H&E images needs to be improved to better highlight Nageotte nodules morphology.
3. I'm curious if there was a group included that had diabetic neuropathy without pain. This section is a bit confusing, as it appears that Figure 1B (quantification) includes a "diabetic with peripheral neuropathy" group, which, based on the text, seems to have a pain component. It might be helpful to improve the group labels for clarity, specifically distinguishing between those with and without pain under diabetes.
4. The authors are encouraged to provide further clarification on the quantification of Nageotte nodule presented in Figure 1D. Was the presence of GFAP/DAPI co-staining sufficient to count them as positive? I appreciate the level of details in the methods section, but I think it can benefit from further clarification to understand what classified as a nodule.
5. What was the control group implemented in the transcriptomics experiments?
6. In addition, it would be helpful to have more details on the clinical characteristics (including analgesics/ or not) of the diabetic donors included in the transcriptomics study.
7. Since the authors initially reported the nodule score increased with DPN, I am curious to see whether the transcriptomic profiles from different patient cohorts (described in figure 1) would differ accordingly. That would give additional insights into how Nageotte nodules might contribute to disease pathogenesis.
8. Did the authors examine any correlations between differentially expressed genes and clinical parameters related to either diabetes or neuropathy?

Reviewer #2

(Remarks to the Author)

Shiers, Price and colleagues provide compelling evidence that Nageotte nodules, dead sensory neurons engulfed by non-neuronal cells and 'innervated' by sprouting axons, occur in human DRG from patients with diabetic peripheral neuropathy, present evidence that these nodules may contribute to pain in patients with diabetic peripheral neuropathy, as well as suggest potential underlying mechanisms. In studies of the composition of Nageotte nodules in DRG from patients with diabetes, their findings support the suggestion that these nodules consist predominantly of satellite glia and non-myelinating Schwann cells (that express SPP1) interspersed with neuronal processes that provide evidence derived from adjacent neuronal soma. Importantly, all experiments were done on human DRG harvested from diabetic and control organ donors, with a substantial number likely having pain and peripheral neuropathy. Their analysis of the cellular composition of

Nageotte nodules, including evidence to exclude innervation by sympathetic postganglionic neurons (as described by McLachlin and Janig following peripheral nerve injury), as well as compelling evidence for innervation by sprouting from adjacent neurons, provides compelling evidence in support of neuronal/non-neuronal interactions in the generation of chronic pain in diabetic painful peripheral neuropathy. Finally, these investigators have employed spatial transcriptomics to profile Nageotte nodules and adjacent DRG neurons, to elucidate potential interactions between Nageotte nodules and these adjacent neuronal soma, which may contribute to the sprouting processes that 'innervate' the nodules, they observed 5 clusters, including one, cluster 3, that is enriched in actively sprouting nociceptor axons harboring mRNAs that might be locally translated, a phenomenon that this group has strongly implicated as playing a critical role in the transition from acute to chronic pain. Taken together, their findings elegantly support the suggestion that signaling from Nageotte nodules to sensory neurons in dorsal root ganglia of patients with diabetes provide novel mechanisms that may influence the excitability of sensory neurons in patients with diabetic painful peripheral neuropathy. Their findings provide novel insight into candidate mechanisms underlying pain in patients with diabetic neuropathy, opening the possibility that neuroprotective interactions that treat neurodegeneration could be a novel class of treatments for diabetic painful peripheral neuropathy and, potentially, other forms of neuropathic pain.

Minor concerns:

There is a minor concern with respect to how well these investigators were able to establish that a given patient had painful diabetic neuropathy. While for an individual patient this may be somewhat problematic, averaging over all patients with diabetes, in their study, many of whom likely have painful peripheral neuropathy, it is highly likely that while they cannot distinguish nodules from patients with pain from those without, their conclusions are reasonable and appropriate. Also, they suggest the presence of painful diabetic neuropathy based on consumption of drugs used to treat pain in patients with diabetic neuropathy. As they are also used to treat other pain syndromes, recognition of this caveat may be indicated.

Do they have any idea of the population of some size distribution contributing to Nageotte nodules. Since it is 25% of neurons, they could look at size distribution of remaining neurons and compare to control DRG. However, this may be beyond the scope of the present study.

While their data suggest Nageotte nodules contribute to neuropathic pain in patients with diabetes, they still don't know if Nageotte nodules are sufficient to produce neuropathic pain, a point that could use some discussion.

How good are histological antibodies? Controls for immuno?

Reviewer #3

(Remarks to the Author)

Summary

The current manuscript investigates the molecular landscape surrounding Nageotte nodules in the dorsal root ganglia of 90 organ donors, 19 of whom had diabetic painful neuropathy (DPN). DPN is common form of neuropathic pain and is associated with spontaneous stabbing pain due to the loss of sensory axons. The authors here seek to advance our current understanding of this process using high resolution microscopy with immunohistochemistry and spatial transcriptomics to elucidate the molecular landscape surrounding Nageotte nodules. A noted weakness of the paper was the choice to perform Visium on only diabetic donors with neuropathic pain, rather than contrasting diagnostic groups. Still, this is a strong paper that has the potential to open new avenues for developing novel therapeutics. The manuscript is well-written and the methods are sound. Clarity around the spatial transcriptomic methods would help strengthen the manuscript. Finally, I would encourage authors to consider making all code available upon submission for reviewers to access.

Major comments

It's unclear from the main text and figure how many donors and sections were used for Visium and what the diagnosis status was for these data. I suggest the authors consider both including this in the results as well as a schematic in the Figure 4. It appears that all 16 Visium samples were from diabetic donors exhibiting neuropathic pain. A more interesting design would have compared at least two diagnostic groups. While this information can be found in sTable 1, the manuscript would benefit from this information being readily available in the main text.

It's unclear from the methods or main text how the spots with Nageotte nodules were found and annotated. Was this done manually or through an automated method with image segmentation? Fig4A is a nice example of how the spots were chosen, but it would be nice to see the colored annotated of Nageotte nodules, Nearby neurons, and Other across 16 samples in supplementary file. In addition, the author should show at least one example Visium section with side-by-side comparisons of manual annotations and established marker genes.

While the authors convincingly show that the Nageotte clusters are not driven by donors, they have not shown if they might be driven by any other batch effect such as a slide. It would be worthwhile to include another UMAP in sFig colored by individual Visium slide to counter this.

Minor comments

- Figure 4F and sFigure 9: It's unclear what the reduced dimensional space being shown is.

- Fig4G: The title "Source of mRNA transcripts in clusters" is strong wording for deconvolution results. Many benchmarks have now shown deconvolution to be a solid estimate at best.

Version 1:

Reviewer comments:

Reviewer #1

(Remarks to the Author)

I have no additional comments, as my previous points have been addressed.

(Remarks on code availability)

For some reason, I was not able to view it.

Reviewer #2

(Remarks to the Author)

The authors have provided thorough and complete responses to concerns raised during the initial review of this manuscript, which is now considered acceptable for publication.

(Remarks on code availability)

Reviewer #3

(Remarks to the Author)

The authors have sufficiently satisfied all of the concerns raised by the referees.

(Remarks on code availability)

The DOI provided for the code does not currently link to anything.

Response to Reviewers

Reviewer #1 (Remarks to the Author):

In this study, the authors used elegant spatial sequencing and histology approaches to uncover the cellular and molecular nature of Nageotte nodules from DRG tissue of organ donors with DPN. They found that these structures primarily consist of satellite glial cells and Schwann cells and that they may contribute to painful neuropathy pathogenesis by regulating axonal sprouting and hyperexcitability. The paper presents a novel and timely contribution to the DPN field, addressing pain and sensory loss, which are currently understudied. However, I have a few comments that I believe should be addressed to enhance the clarity and rigor of the work:

1. Currently, the introduction treats Nageotte nodules and diabetic neuropathy pathogenesis as two separate topics, which may make it difficult for readers to appreciate their interrelationship, which is a main theme of the study. Consequently, I recommend better integrating the discussion of how elucidating the role of Nageottes nodules could enhance our understanding of painful diabetic neuropathy. This connection will improve introduction clarity and provide a stronger context for the readers.

Thank you for your feedback, we completely agree. We have now added additional text to both the Introduction and Discussion sections (highlighted in blue) which better integrate the relationship between Nageotte nodules and diabetic peripheral neuropathy. We hope this improves clarity for our readers.

2. The quality of H&E images needs to be improved to better highlight Nageotte nodules morphology.

We have now added three panels (**Fig 1A**) showing a human DRG stained with Hematoxylin and Eosin with increasing magnification to demonstrate the morphology of Nageotte nodules.

3. I'm curious if there was a group included that had diabetic neuropathy without pain. This section is a bit confusing, as it appears that Figure 1B (quantification) includes a "diabetic with peripheral neuropathy" group, which, based on the text, seems to have a pain component. It might be helpful to improve the group labels for clarity, specifically distinguishing between those with and without pain under diabetes.

The organ donor medical histories are summaries of hospital records and information from family members which we receive from the organ procurement organization. Diabetes diagnosis is known since the donor's insulin and blood sugar are monitored and maintained while on life support. However, details about pain and neuropathy can sometimes be vague or not included (or undiagnosed) so we had to make inferences based on a number of keywords in the donor's medical summaries. These details are found in the methods section, and the keywords are bolded in the donor's medical histories found in **Supplemental Table 1**.

We have included the following paragraphs in the results section about the donor groupings and about medical histories to make this clearer in the main text:

"We first conducted Hematoxylin and Eosin staining on all DRGs (90 donors) to visualize Nageotte nodules which appear as dense clusters of non-neuronal nuclei (**Fig 1A**). Each DRG was qualitatively scored for the presence of Nageotte nodules using a 5-point scaling system ranked from very low-to-high (**Fig 1B**). Following scoring, donors were categorized into groups based on keywords in their medical histories (bolded in **Supplementary Table 1**). The "diabetic with peripheral neuropathy" group included diabetic donors with documented neuropathy, nerve pain, and/or amputation. The "diabetic taking analgesics" group consisted of diabetic donors without neuropathy-related keywords but who were prescribed analgesics (gabapentin,

duloxetine, hydrocodone, etc). The “diabetic not taking analgesics” group includes diabetics with no neuropathy-related keywords and no prescribed analgesics. The “no diabetes” group consisted of non-diabetic donors without DRG-related chronic pain conditions such as arthritis, neuropathy, or fibromyalgia. The results showed that Nageotte nodules were significantly more prevalent in diabetics with peripheral neuropathy, diabetics taking analgesics, and diabetics not taking analgesics compared to non-diabetic controls (**Fig 1C**), respectively.

It is important to note that the donor medical histories are provided by the organ procurement organization (OPO) and are summaries of hospital records and family-reported information. In some cases, neuropathy or pain may have been undiagnosed, or unreported by the donor’s family. For example, donor #12 who had high Nageotte nodule content, was taking gabapentin and duloxetine (two widely prescribed analgesics for diabetic nerve pain), and had difficulty walking for the past 6 weeks fell under the category of “diabetic taking analgesics.” However, this donor likely had neuropathy, but it was not explicitly documented in the medical history summary, so it did not meet the criteria for being grouped as “diabetic with peripheral neuropathy.”

4. The authors are encouraged to provide further clarification on the quantification of Nageotte nodule presented in Figure 1D. Was the presence of GFAP/DAPI co-staining sufficient to count them as positive? I appreciate the level of details in the methods section, but I think it can benefit from further clarification to understand what classified as a nodule.

Similar to Hematoxylin, a nuclear stain like DAPI can be used to distinguish Nageotte nodules as has been reported in previous pathology reports and case studies¹⁻⁹. However, we found that adding GFAP staining made the identification of Nageotte nodules more obvious as GFAP strongly labeled the cells at Nageotte nodules as shown in **Fig 1D**. Nageotte nodules were also absent of a peripherin-positive neuronal soma. We have added this detail to the methods section.

5. What was the control group implemented in the transcriptomics experiments?

Finding an appropriate control for the Nageotte nodule analysis was challenging as Nageotte nodules are pathological structures that form after a sensory neuron dies. As such, there is no “healthy” Nageotte nodule for differential expression analysis. If we compared the Nageotte nodule barcodes to neuronal barcodes (which overlap the neural satellite glia as well), then all of the neuronally enriched genes would be differentially expressed (more expressed in the neurons) and we would lose the axonally trafficked mRNA signature that is present in Nageotte arborizations. Similarly, since the neuronal barcodes also overlap satellite glia, the glial signature would also show up differentially expressed, making it difficult to pinpoint the cell type present at Nageotte nodules and what genes are important within their transcriptomic signatures.

While we did run non-diabetic controls in the spatial transcriptomics experiments, we did not use these data for the purposes of Nageotte nodule characterization. Instead, we pseudo-bulked Nageotte nodule barcodes from 6 DPN donors and looked at enrichment of genes within these structures using a ranking analysis. We completed validation experiments using RNAscope *in situ* hybridization and IHC to confirm several findings such as *SPP1*, *CD44*, *SOX10*, *S100*, *CD68*, *FABP7* which represent expression of genes that were enriched in Nageotte nodules and specific cell type markers for cells identified by deconvolution. We also used single-cell data on human DRG to deconvolute the transcriptomic signature of Nageotte nodules and provide estimations on what cell types comprise these structures.

6. In addition, it would be helpful to have more details on the clinical characteristics (including analgesics/ or not) of the diabetic donors included in the transcriptomics study.

The medical history report for each donor is included in Supplemental Table 1.

7. Since the authors initially reported the nodule score increased with DPN, I am curious to see whether the transcriptomic profiles from different patient cohorts (described in figure 1) would differ accordingly. That would give additional insights into how Nageotte nodules might contribute to disease pathogenesis.

Thank you for your comment, we agree this would be interesting to pursue but may be better addressed in future experiments. In this paper, our goal was to characterize the anatomical and transcriptomic profiles of Nageotte nodules since there is limited information on these pathological structures. As such, we only used DPN donors for the spatial transcriptomic experiments and did not sample from other diabetic donors with different medical history profiles.

However, we are especially interested in disease pathogenesis and think that future experiments looking at Nageotte nodule formation *in vitro* will provide insights into how these structures form and could help us to address if the cells forming Nageotte nodules change over time or across disease progression. We do not currently have any evidence that the cells forming Nageotte nodules change phenotypes across donors, only that they are more prevalent in individuals with neuropathy, especially in conditions involving neurodegeneration/neuronal loss. However, we found that there are different types of Nageotte nodules based on the clustering results from the sequencing data, so it is possible that one structural type is more prevalent in early stages of neurodegeneration in diabetes. There is a lot to do, and we are very excited about what these structures could reveal in regard to pain processing and treatments for neuropathy.

8. Did the authors examine any correlations between differentially expressed genes and clinical parameters related to either diabetes or neuropathy?

No, for the purposes of this paper, we only analyzed spatial transcriptomic data of Nageotte nodule barcodes from 6 DPN samples. All of these donors had diabetes and keywords indicating neuropathy from their medical history as shown in Supplemental Table 1. We did not sequence donors with other clinical parameters.

Reviewer #2 (Remarks to the Author):

Shiers, Price and colleagues provide compelling evidence that Nageotte nodules, dead sensory neurons engulfed by non-neuronal cells and ‘innervated’ by sprouting axons, occur in human DRG from patients with diabetic peripheral neuropathy, present evidence that these nodules may contribute to pain in patients with diabetic peripheral neuropathy, as well as suggest potential underlying mechanisms. In studies of the composition of Nageotte nodules in DRG from patients with diabetes, their findings support the suggestion that these nodules consist predominantly of satellite glia and non-myelinating Schwann cells (that express SPP1) interspersed with neuronal processes that provide evidence derived from adjacent neuronal soma. Importantly, all experiments were done on human DRG harvested from diabetic and control organ donors, with a substantial number likely having pain and peripheral neuropathy. Their analysis of the cellular composition of Nageotte nodules, including evidence to exclude innervation by sympathetic postganglionic neurons (as described by McLachlin and Janig following peripheral nerve injury), as well as compelling evidence for innervation by sprouting from adjacent neurons, provides compelling evidence in support of neuronal/non-neuronal interactions in the generation of chronic pain in diabetic painful peripheral neuropathy. Finally, these investigators have employed spatial transcriptomics to profile Nageotte nodules and adjacent DRG neurons, to elucidate potential interactions between Nageotte nodules and these adjacent neuronal soma, which may contribute to the sprouting processes that ‘innervate’ the nodules, they observed 5 clusters, including one, cluster 3, that is enriched in actively sprouting nociceptor axons harboring mRNAs that might be locally translated, a phenomenon that this group has strongly implicated as playing a critical role in the transition

from acute to chronic pain. Taken together, their findings elegantly support the suggestion that signaling from Nageotte nodules to sensory neurons in dorsal root ganglia of patients with diabetes provide novel mechanisms that may influence the excitability of sensory neurons in patients with diabetic painful peripheral neuropathy. Their findings provide novel insight into candidate mechanisms underlying pain in patients with diabetic neuropathy, opening the possibility that neuroprotective interactions that treat neurodegeneration could be a novel class of treatments for diabetic painful peripheral neuropathy and, potentially, other forms of neuropathic pain.

Minor concerns:

1. There is a minor concern with respect to how well these investigators were able to establish a that a given patient had painful diabetic neuropathy. While for an individual patient this may be somewhat problematic, averaging over all patients with diabetes, in their study, many of whom likely have painful peripheral neuropathy, it is highly likely that while they cannot distinguish nodules from patients with pain from those without, their conclusions are reasonable and appropriate. Also, they suggest the presence of painful diabetic neuropathy based on consumption of drugs used to treat pain in patients with diabetic neuropathy. As they are also used to treat other pain syndromes, recognition of this caveat may be indicated.

We thank you for your review and completely agree with your statement. Donor medical information is provided by the OPO and is a summary of medical information from hospital records and next of kin. We used keywords in the medical summaries to make inferences about pain (bolded in **Supplemental Table 1**), but in many cases we do not know if the analgesic was used for diabetic neuropathic pain. In revising our work, we have changed the wording to indicate “diabetic peripheral neuropathy” instead of “diabetic painful neuropathy.” We also have included more details in the results and methods sections about donor groupings and their medical histories.

2. Do they have any idea of the population of some size distribution contributing to Nageotte nodules. Since it is 25% of neurons, they could look at size distribution of remaining neurons and compare to control DRG. However, this may be beyond the scope of the present study.

We think this is a great idea. We have added an additional Supplemental Figure that contains two new datasets that address this question.

In **Supplemental Figure 2A-B**, we measured the diameter of neurons with visible nuclei in non-diabetic and DPN DRGs. We grouped these neurons as small, medium, or large based on size classifications for human C-nociceptors, A δ neurons, and A β neurons which we previously described in our spatial transcriptomic publication on human DRG¹⁰. When calculated as a fraction of the entire population (neurons with visible nuclei + Nageotte nodules), we observed a significant decrease in small diameter neurons in the DPN DRGs.

In **Supplemental Figure 2C-D**, we counted the number of *SCN10A* (Nav1.8) positive and negative neurons in non-diabetic and DPN DRGs. We found a significant decrease in the percentage of *SCN10A*+ neurons in DPN DRGs.

3. While their data suggest Nageotte nodules contribute to neuropathic pain in patients with diabetes, they still don't know if Nageotte nodules are sufficient to produce neuropathic pain, a point that could use some discussion.

We agree completely. We have added text to the discussion section on our hypotheses surrounding Nageotte nodules and pain signaling, and some ideas for future experiments.

4. How good are histological antibodies? Controls for immuno?

The antibodies used for all histology experiments and notes about their specificity are found in **Supplemental Table 5**. We used knock-out validated, monoclonal antibodies whenever possible. We have used many of these antibodies (peripherin, Nav1.7, TH, and GFAP) extensively on human tissues, particularly human DRG, TG, and spinal cord and they have given the predicted pattern in regard to subcellular, cellular, and regional expression.

Negative controls were run for all samples in every immunofluorescence experiment which were exposed to all reagents except for primary antibody. We used a post-staining solution called True Black to quench a lot of the autofluorescent lipofuscin present in human neurons which tremendously decreased background noise in our experiments. For the negative controls, unremarkable background autofluorescence was noted particularly in the 488 (green) channel, but otherwise all channels were virtually void of signal. We have added some representative negative control images in **Supplemental Figure 14**. These two images represent extremes, where we can see virtually no background labeling in one donor, while some unsubstantial green autofluorescence labeling in the fibers from another. Details about the negative controls, True Black, and lipofuscin can be found in the methods section.

Reviewer #3 (Remarks to the Author):

Summary

The current manuscript investigates the molecular landscape surrounding Nageotte nodules in the dorsal root ganglia of 90 organ donors, 19 of whom had diabetic painful neuropathy (DPN). DPN is common form of neuropathic pain and is associated with spontaneous stabbing pain due to the loss of sensory axons. The authors here seek to advance our current understanding of this process using high resolution microscopy with immunohistochemistry and spatial transcriptomics to elucidate the molecular landscape surrounding Nageotte nodules. A noted weakness of the paper was the choice to perform Visium on only diabetic donors with neuropathic pain, rather than contrasting diagnostic groups. Still, this is a strong paper that has the potential to open new avenues for developing novel therapeutics. The manuscript is well-written and the methods are sound. Clarity around the spatial transcriptomic methods would help strengthen the manuscript. Finally, I would encourage authors to consider making all code available upon submission for reviewers to access.

Major comments

1. It's unclear from the main text and figure how many donors and sections were used for Visium and what the diagnosis status was for these data. I suggest the authors consider both including this in the results as well as a schematic in the Figure 4. It appears that all 16 Visium samples were from diabetic donors exhibiting neuropathic pain. A more interesting design would have compared at least two diagnostic groups. While this information can be found in STable 1, the manuscript would benefit from this information being readily available in the main text.

Thank you for your comment. We have added new text to the methods and results sections and to **Figure 4A** which provides information about the number of sections and donors that were used in spatial transcriptomic experiments.

All of the samples used in the Nageotte nodule VISIUM analyses were DPN donors. While we think the idea of comparing Nageotte nodules across diabetic donor groupings is a compelling idea, we believe that work is better suited for a future paper, and for a different spatial transcriptomic technology using markers we identified in this work. In our current work, our mission was to anatomically and molecularly characterize these pathological structures as there is very limited information on them. For instance, the available literature indicates that Nageotte nodules are comprised of satellite glial cells, but those claims originated from 100-year-old studies that used Golgi staining and not molecular markers. We do not currently have any evidence that the cells forming Nageotte nodules change phenotypes across donors, only that they are more prevalent in individuals with

neuropathy, or in conditions involving neurodegeneration/neuronal loss. However, we found that there are different types of Nageotte nodules based on the clustering results from the sequencing data, so it is possible that one structural type is more prevalent in early stages of neurodegeneration in diabetes. There is a lot to do, and we are very excited about what these structures could reveal in regard to pain processing and treatments for neuropathy.

Finding an appropriate control for the Nageotte nodule analysis was challenging as Nageotte nodules are pathological structures that form after a sensory neuron dies. As such, there is no “healthy” Nageotte nodule for differential expression analysis. Comparing the structures to either non-diabetic or DPN neuronal barcodes (which also overlap with neural satellite glial cells) would result in differentially expressed genes for neurons (higher enrichment in the neuron soma) which would mask axonally trafficked mRNAs found in the Nageotte nodule arborizations and may also hide the satellite glial cell signature. As such, we pseudo-bulked all Nageotte nodule barcodes from the DPN donors (6 donors) and used our recently published single-nuclei RNA sequencing dataset on human DRG to assess gene and potential cell-type enrichment at Nageotte nodules. These details are found in the methods section.

We realize there are limitations to this approach, especially without a “healthy” biological control, so we plan to use alternative methods such as *in situ* transcriptomics, likely the Xenium technology, in our future work. This approach will allow us to look at gene expression in single nuclei at Nageotte nodules and then be able to compare the cell types to their counterparts in the DPN samples and within non-diabetic controls. From this perspective, it is important to note that our current work enables that future work as the discovery-based transcriptomics done here establishes panels to be used in future experiments.

2. It’s unclear from the methods or main text how the spots with Nageotte nodules were found and annotated. Was this done manually or through an automated method with image segmentation?

Nuclear stains like hematoxylin can be used to distinguish Nageotte nodules¹⁻⁹, and neurons are easy to visualize in human DRG due to their large cell bodies. As such, we manually selected the barcodes touching Nageotte nodules and nearby neurons in Loupe Browser. We have now added the following text to the methods section:

“VISIUM sections were examined in Loupe Browser (10X Genomics) and barcodes that overlapped Nageotte nodules and nearby neurons were manually selected. Barcodes that overlapped individual or multiple Nageotte nodules were annotated as “Nageotte Nodules.” Barcodes that overlapped Nageotte nodules and neurons were annotated as “Nageotte also touching neuron” and were not included in the analysis. Barcodes touching the soma of a neuron that was near a Nageotte nodule (within 1 barcode) were annotated as “nearby neurons.” All barcodes that did not fit into these categories were considered to be “Other” barcodes. The annotated barcodes with instructions on how to import them into the downloadable Loupe Browser files can be found in **Supplemental Table 6.**”

3. Fig4A is a nice example of how the spots were chosen, but it would be nice to see the colored annotated of Nageotte nodules, Nearby neurons, and Other across 16 samples in supplementary file.

We agree and have tried to satisfy this request in the best possible way. The human DRG sections occupy most of the 6.5mm x 6.5mm etched frame of the VISIUM slides. Therefore, in order to visualize the entire tissue section, we would need to use a low magnification view which would make it difficult to clearly visualize the structures (neurons and Nageotte nodules that are ~50µm in size). To show an example, we have included the H&E image file with and without the annotated barcodes (**Supplemental Figure 9A**), and a zoomed in version of a small area of the section (**Supplemental Figure 9B**).

However, for all sections, we have made the Loupe Browser files publicly available (on the SPARC platform with DOI indicated under data availability) and added a new table (**Supplemental Table 6**) which contains all of the annotated barcodes. The first sheet within this table contains clear instructions on how to install Loupe Browser and import the annotated barcodes onto each tissue section within the program.

4. In addition, the author should show at least one example Visium section with side-by-side comparisons of manual annotations and established marker genes.

We have added the requested images to **Supplemental Figure 9** which now includes the top 10 marker genes for the cell types identified in the deconvolution analysis overlaid on the H&E image for spatial visualization. We also have included a subpanel of marker genes that we found to be expressed in Nageotte nodules such as *SOX10* (confirmed with RNAscope), *SPP1* (confirmed with RNAscope) and *CLU*. All Loupe Browser files can be downloaded (instructions and barcode annotations in **Supplemental Table 6**), and marker genes found in **Supplemental Table 3** can be searched in the program for spatial visualization of all genes on each section.

5. While the authors convincingly show that the Nageotte clusters are not driven by donors, they have not shown if they might be driven by any other batch effect such a slide. It would be worthwhile to include another UMAP in sFig colored by individual Visium slide to counter this.

Thank you for pointing out this oversight. The requested UMAP can now be found in **Supplemental Figure 9D**.

Minor comments

6. Figure 4F and sFigure 9: It's unclear what the reduced dimensional space being shown is.

Umap axes are now included for both **Figure 4F** and **Supplemental Figure 9**.

7. Fig4G: The title "Source of mRNA transcripts in clusters" is strong wording for deconvolution results. Many benchmarks have now shown deconvolution to be a solid estimate at best.

We agree. We have changed the figure title and caption to indicate that it is an estimate: "Estimated source of mRNA transcripts in clusters"

References

- 1 Marshall, A. & Duchon, L. W. Sensory system involvement in infantile spinal muscular atrophy. *J Neurol Sci* **26**, 349-359, doi:10.1016/0022-510x(75)90207-5 (1975).
- 2 Hanani, M. Satellite Glial Cells in Human Disease. *Cells* **13**, doi:10.3390/cells13070566 (2024).
- 3 Sonoda, K. *et al.* TAR DNA-binding protein 43 pathology in a case clinically diagnosed with facial-onset sensory and motor neuronopathy syndrome: an autopsied case report and a review of the literature. *J Neurol Sci* **332**, 148-153, doi:10.1016/j.jns.2013.06.027 (2013).
- 4 Fratkin, J. D., Leis, A. A., Stokic, D. S., Slavinski, S. A. & Geiss, R. W. Spinal cord neuropathology in human West Nile virus infection. *Arch Pathol Lab Med* **128**, 533-537, doi:10.5858/2004-128-533-SCNIHW (2004).
- 5 Scaravilli, F., Giometto, B., Chimelli, L. & Sinclair, E. Macrophages in human sensory ganglia: an immunohistochemical and ultrastructural study. *J Neurocytol* **20**, 609-624, doi:10.1007/BF01215268 (1991).
- 6 Pandya, S. S. & Bhatki, W. S. Severe pan-sensory neuropathy in leprosy. *Int J Lepr Other Mycobact Dis* **62**, 24-31 (1994).

- 7 Makino, M., Hiwatashi, D., Minemura, K. & Kawaguchi, K. Autonomic and sensory ganglionopathy occurring in a patient with fulminant type 1 diabetes mellitus. *Pathol Int* **66**, 102-107, doi:10.1111/pin.12373 (2016).
- 8 Yoshioka, M. *et al.* Expression of HIV-1 and interleukin-6 in lumbosacral dorsal root ganglia of patients with AIDS. *Neurology* **44**, 1120-1130, doi:10.1212/wnl.44.6.1120 (1994).
- 9 Krarup-Hansen, A. *et al.* Histology and platinum content of sensory ganglia and sural nerves in patients treated with cisplatin and carboplatin: an autopsy study. *Neuropathol Appl Neurobiol* **25**, 29-40, doi:10.1046/j.1365-2990.1999.00160.x (1999).
- 10 Tavares-Ferreira, D. *et al.* Spatial transcriptomics of dorsal root ganglia identifies molecular signatures of human nociceptors. *Sci Transl Med* **14**, eabj8186, doi:10.1126/scitranslmed.abj8186 (2022).